# Region-Specific Homeostatic Identity of Astrocytes Is Essential for Defining Their Response to Pathological Insults

**DOI:** 10.3390/cells12172172

**Published:** 2023-08-30

**Authors:** Natallia Makarava, Olga Mychko, Kara Molesworth, Jennifer Chen-Yu Chang, Rebecca J. Henry, Natalya Tsymbalyuk, Volodymyr Gerzanich, J. Marc Simard, David J. Loane, Ilia V. Baskakov

**Affiliations:** 1Center for Biomedical Engineering and Technology, University of Maryland School of Medicine, Baltimore, MD 21201, USA; nmakarava@som.umaryland.edu (N.M.); cchang1@som.umaryland.edu (J.C.-Y.C.); 2Department of Anatomy and Neurobiology, University of Maryland School of Medicine, Baltimore, MD 21201, USA; 3Department of Anesthesiology and Shock, Trauma and Anesthesiology Research (STAR) Center, University of Maryland School of Medicine, Baltimore, MD 21201, USAloanedj@tcd.ie (D.J.L.); 4Department of Neurosurgery, University of Maryland School of Medicine, Baltimore, MD 21201, USA; 5Department of Pathology, University of Maryland School of Medicine, Baltimore, MD 21201, USA; 6Department of Physiology, University of Maryland School of Medicine, Baltimore, MD 21201, USA; 7School of Biochemistry and Immunology, Trinity Biomedical Sciences Institute, Trinity College Dublin, D02 R590 Dublin, Ireland

**Keywords:** astrocytes reactivity, neurodegenerative diseases, prion disease, Alzheimer’s disease, traumatic brain injury, aging

## Abstract

The transformation of astrocytes into reactive states constitutes a biological response of the central nervous system under a variety of pathological insults. Astrocytes display diverse homeostatic identities that are developmentally predetermined and regionally specified. Upon transformation into reactive states associated with neurodegenerative diseases and other neurological disorders, astrocytes acquire diverse reactive phenotypes. However, it is not clear whether their reactive phenotypes are dictated by region-specific homeostatic identity or by the nature of an insult. To address this question, region-specific gene expression profiling was performed for four brain regions (cortex, hippocampus, thalamus, and hypothalamus) in mice using a custom NanoString panel consisting of selected sets of genes associated with astrocyte functions and their reactivity for five conditions: prion disease, traumatic brain injury, brain ischemia, 5XFAD Alzheimer’s disease model and normal aging. Upon transformation into reactive states, genes that are predominantly associated with astrocytes were found to respond to insults in a region-specific manner. Regardless of the nature of the insult or the insult-specificity of astrocyte response, strong correlations between undirected GSA (gene set analysis) scores reporting on astrocyte reactivity and on their homeostatic functions were observed within each individual brain region. The insult-specific gene expression signatures did not separate well from each other and instead partially overlapped, forming continuums. The current study demonstrates that region-specific homeostatic identities of astrocytes are important for defining their response to pathological insults. Within region-specific populations, reactive astrocytes show continuums of gene expression signatures, partially overlapping between individual insults.

## 1. Introduction

The transformation of astrocytes into reactive states constitutes a biological response of the central nervous system (CNS) under a variety of pathological conditions, including traumatic brain injury (TBI), ischemic stroke, brain infection, and neurodegenerative diseases such as Alzheimer’s, Parkinson’s, and prion diseases [1,2]. Astrocytes’ ability to respond to a variety of pathological conditions is one among their many other vital functions. These functions include supporting neuronal growth, modulating neurotransmission, forming and maintaining synapses, regulating blood flow, maintaining the blood–brain barrier (BBB), and providing energy and metabolic support to neurons [3,4,5]. It remains unclear whether, in their reactive state, these critical astrocytic functions become universally affected, and whether transitioning into a reactive state contributes to brain impairments or promotes neuroprotection and neuronal repair.

Studies that employed imaging and physiological and transcriptome approaches revealed that, under normal conditions, astrocytes are functionally diverse and form distinct neural-circuit-specialized subpopulations [6,7,8,9,10,11,12,13]. In fact, seven developmentally predetermined sub-types of astrocytes that reside in different brain regions have been identified in mouse brains [10]. Upon transformation into reactive states associated with neurodegenerative diseases and other neurological disorders, astrocytes also acquire diverse and sometimes mixed phenotypes [14,15,16,17,18,19,20]. However, the role of a region-specific homeostatic identity of astrocytes versus the nature of an insult in defining their reactive phenotypes is not well understood. Do astrocytes respond to pathological insults in an insult-specific manner, adopting insult-specific states? If so, are the insult-specific states uniform across brain regions? Investigating the role of the homeostatic identity of astrocytes versus the nature of the pathological insult in driving their reactive phenotypes is important for learning how astrocyte reactivity can be modulated to alleviate pathology.

In order to define the role of regional identity versus the nature of insult, the current study performed gene expression profiling under five conditions: prion disease, TBI, brain ischemia, 5XFAD Alzheimer’s disease (AD) model [21], and normal aging (24-month-old mice). We opted for a targeted approach and a bulk tissue analysis using a custom NanoString panel consisting of selected sets of genes that report on astrocyte functions and their reactivity. Data were collected from four brain region: cortex, hippocampus, thalamus, and hypothalamus. We found that region-specific homeostatic identities of astrocytes define their gene expression signatures under pathological insults. Within individual brain regions, continuums of gene expression signatures partially overlap between individual insults.

## 2. Materials and Methods

### 2.1. Ethics Approval and Consent to Participate

The study was carried out in accordance with the recommendations in the Guide for the Care and Use of Laboratory Animals of the National Institutes of Health. The animal protocol was approved by the Institutional Animal Care and Use Committee of the University of Maryland, Baltimore (Assurance Number: A32000-01; Protocol Number: 0118001).

### 2.2. Key Resources 

Key resources are listed in Appendix A.

### 2.3. Prion Infection

Using isoflurane anesthesia, 6-week-old C57Bl/6J female and male mice were subjected to intraperitoneal inoculation of 1% brain homogenate, prepared in PBS (pH 7.4) using terminally ill SSLOW-infected mice (originally abbreviated as Synthetic Strain Leading to OverWeight) [22,23]. Brain materials from the 5th passage of SSLOW in mice was used for inoculations [24]. Inoculation volume was 200 μL. 1× PBS was inoculated into normal control groups (Norm). Animals were regularly observed for signs of neurological impairment: abnormal gait, hind limb clasping, lethargy, and weight loss. Mice were considered terminally ill and were euthanized when they were unable to rear and/or lost 20% of their weight (at 166–185 dpi). 

### 2.4. Traumatic Brain Injury

Traumatic brain injury was administered as a controlled cortical impact (CCI) using a custom-designed CCI device, consisting of a microprocessor-controlled pneumatic impactor with a 3.5 mm diameter tip, as described [25]. Briefly, mice were anesthetized with isoflurane administered through a nose mask and placed on a heated pad to maintain their core body temperature at 37 °C. The head was mounted in a stereotaxic frame, a 10-mm midline incision was made over the skull and the skin and fascia were reflected. A 5-mm craniotomy was made on the central aspect of the left temporoparietal bone, between the bregma and lambda. The impounder tip of the injury device was then extended to its full stroke distance (44 mm), positioned to the surface of the exposed dura, and reset to impact the cortical surface. Moderate-level CCI was induced using an impactor velocity of 6 m/s, with a deformation depth of 2 mm, as described [25]. After injury, the incision was closed with interrupted 6-0 silk sutures, anesthesia was terminated, and the animal was placed into a heated chamber to maintain normal core temperature for 45 min post-injury. Control mice underwent sham procedure consisting of anesthesia, skin reflection, and suture, but did not receive the craniotomy or impact. Mice were monitored daily post-injury. 

### 2.5. Transient Middle Cerebral Artery Occlusion (MCAO)

The procedure for MCAO in mice has been described in detail [26]. Briefly, mice (20–25 gm, 8–12 weeks) were anesthetized (induction, 3.0% isoflurane; maintenance, 1.5–2.0% isoflurane with a mixture of O_2_ (200 mL/min) and N_2_O (800 mL/min)). Body temperature was maintained during surgery (36.5 ± 0.5 °C) with a feedback heating-controlled pad system (Harvard Apparatus, Holliston, MA, USA). A midline ventral neck incision was made, and the left common carotid artery (CCA) was ligated proximal to the bifurcation. A second suture was placed around the CCA near the bifurcation. A small arteriotomy was made in the CCA between two sutures. A silicon filament (602356PK5Re Doccol Corp., Redlands, CA, USA) was introduced into the CCA and advanced into the ICA, ~8.0 mm from bifurcation until it occludes the MCA, where resistance is felt. The silicon filament was secured by sutures and was left in place for 2 h. Neurological behavior was monitored, and any animal that did not show circling behavior was excluded. After 2 h, the animal was re-anesthetized, the occluder filament was removed, and the CCA was ligated. Sham treatment of the control group included anesthesia and surgical incision in the neck but no middle cerebral artery occlusion.

### 2.6. 5XFAD Mouse Model of Alzheimer’s Disease

Male and female 5XFAD mice (B6SJL-Tg(APPSwFlLon, PSEN1*M146L*L286V) 6799Vas/Mmjax, Jax.org catalog # 34840) and WT littermates were group-housed together in random ratios, three to five mice per cage. All pups were genotyped using Transnetyx genotypic services (Cordova, TN, USA). All 5XFAD transgenic mice were hemizygous with respect to the transgene. All mice were kept on a 12 h light/dark cycle. 

### 2.7. Brain Tissue Collection and RNA Isolation

At the designated time points (Table 1), mice were euthanized by CO_2_ asphyxiation and their brains were immediately extracted. The brains were kept ice-cold for prompt dissection or preserved in 10% buffered formalin (MilliporeSigma, Burlington, MA, USA) for histopathology.

Extracted ice-cold brains were dissected using a rodent brain slicer matrix (Zivic Instruments, Pittsburg, PA, USA). A 2-mm central coronal section of each brain was used to collect individual regions. Allen Brain Atlas digital portal (http://mouse.brain-map.org/static/atlas (accessed on 14 September 2017)) was used as a reference. The hypothalami (HTh), as well as thalami (Th), hippocampi (Hp), and cortices (Ctx) were collected into RNase-free, sterile tubes, frozen in liquid nitrogen, and stored at −80 °C until RNA isolation with an Aurum Total RNA Mini Kit (Bio-Rad, Hercules, CA, USA), as described [27]. Briefly, brain tissue samples were homogenized within RNase-free 1.5-mL tubes in 200 μL of Trizol (Thermo Fisher Scientific, Waltham, MA, USA), using RNase-free disposable pestles (Fisher scientific, Hampton, NH, USA). After homogenization, an additional 600 μL of Trizol was added to each homogenate, and the samples were centrifuged at 11,400× *g* for 5 min at 4 °C. The supernatant was collected, incubated for 5 min at room temperature, then supplemented with 160 μL of cold chloroform and vigorously shaken for 30 s by hand. After an additional 5-min incubation at room temperature, the samples were centrifuged at 11,400× *g* for 15 min at 4 °C. The top layer was transferred to new RNase-free tubes and mixed with an equal amount of 70% ethanol. Subsequent steps were performed using an Aurum Total RNA Mini Kit (Bio-Rad, Hercules, CA, USA) following the manufacturer instructions. Isolated total RNA was subjected to DNase I digestion. RNA purity and concentrations were estimated using a NanoDrop One Spectrophotometer (Thermo Fisher Scientific, Waltham, MA, USA). 

### 2.8. Design of NanoString nCounter Mouse Astrocyte Panel

To design a custom-based nCounter Mouse Astrocyte Panel, we utilized the compilation of astrocyte-enriched genes published by Network Glia, accessible at https://www.networkglia.eu/en/astrocyte (accessed on 7 May 2018). To ascertain their astrocyte-specificity, we cross-referenced these genes with a publicly available database www.brainrnaseq.org (accessed on 7 May 2018). Two hundred seventy-five genes that, under normal conditions, express primarily in astrocytes were selected. For the validation of functional pathways, an in-house literature search was conducted. In addition, 47 genes reporting on reactive phenotypes, including A1-, A2-, pan-specific markers, originally identified by Ben Barres and coauthors [16], and other markers of reactive astrocytes identified via an in-house literature search were used for the panel. The majority of A1, A2, and pan-specific markers lack astrocyte-specificity. Nevertheless, they were included in the panel, because they have been widely utilized within the field for an extended period to monitor the nature of astrocyte reactivity. In addition, 8 microglia (*Aif1*, *CD68*, *Il1a*, *P2ry12*, *Rgs10*, *TLR2*, *TMEM119*, *Tnf*), 10 neuron (*Arc*, *Gabrg1*, *Grin1*, *Gin2b*, *Grm2*, *Nos1*, *Slc32a1*, *Snap25*, *Syn2*, *Syp*), and 2 oligodendrocyte (*Cldn11*, *Sirt2*) -specific genes were included in the panel. The inclusion of 10 housekeeping genes completed the panel, bringing the total number of genes to 352. Recognizing the constraint in the number of genes accommodated within the panels, a choice was made not to create an exhaustive listing of pathways encompassing all astrocyte functions. In certain instances, gene sets containing a limited number of genes were dissolved and their constituent genes were reallocated among other gene sets. Three hundred forty-two genes were assigned to 23 gene sets to allow for an advanced analysis of their group changes (Appendix A). The detection probes were designed by Nanosting Thechnologies to target the maximum number of validated transcript variants and minimize cross-reactivity with related genes or pseudogenes.

### 2.9. Analysis of Gene Expression by NanoString

An amount of 200 ng of total RNA was submitted to the Institute for Genome Sciences at the University of Maryland, School of Medicine, for RNA integrity check and subsequent analysis using a custom nCounter Mouse Astrocyte Panel. Only samples with an RNA integrity number RIN > 7.2 were used for NanoString analysis. For each sample, the assessment of all target sequences was performed within a single tube, using uniquely coded hybridization probes, enabling a reliable and reproducible assessment of the expression. Since the technology does not require an amplification and thus avoids amplification bias, each sample was analyzed once. Each animal group consisted of at least 3 individual samples analyzed in parallel. All data passed quality control assessments for imaging, binding, positive control, or CodeSet content normalization. The analysis of data was performed using nSolver Analysis Software 4.0, including nCounter Advanced Analysis (version 2.0.115) for principal component analysis (PCA) and differential expression analysis. For agglomerative clustering and heat maps, genes with less than 10% of samples above 20 counts were excluded. Z-score transformation was performed for genes. Clustering was done using Euclidian distance, and the linkage method was Average.

Differentially expressed genes (DEGs) were calculated using NanoString nCounter Advanced Analysis, for each brain region separately. Each experimental group was compared to the corresponding control group with all other groups present in the analysis. Only genes with adjusted *p* < 0.1 and linear fold change ≥±1.2 were counted.

Gene set analysis scores (GSA scores) were undirected global significance scores from the gene set analysis of NanoString nCounter, which measured the cumulative evidence for the differential expression of genes in a gene set and were calculated as the square root of mean squared t-statistics of genes using built-in algorithms. Being undirected, GSA scores reported on the tendency of a gene set to have differentially expressed genes and did not take into account if the genes were over- or under-expressed. Each brain region was analyzed separately. To characterize the relationship between astrocyte reactivity and function, the GSA scores of corresponding gene sets were summed and plotted against each other, or further normalized by control group GSAs and plotted. 

### 2.10. Histopathology and Immunofluorescence

Formalin-fixed prion-infected brains and mock-inoculated controls were treated for 1 h with 95% formic acid to deactivate prion infectivity before being embedded in paraffin. All other brains were embedded without formic acid treatment. Subsequently, 4-µm brain sections produced using a Leica RM2235 microtome were mounted on slides and processed for immunohistochemistry. To expose epitopes, slides were subjected to 20 min of hydrated autoclaving at 121 °C in Antigen Retriever citrate buffer, pH 6.0 (C9999, MilliporeSigma). For co-immunofluorescence, rabbit polyclonal anti-Iba1 antibody was used in combination with chicken polyclonal anti-GFAP antibody. The secondary antibodies were goat anti-rabbit or anti-chicken IgG conjugated with Alexa Fluor 546 for red color or Alexa Fluor 488 for green color (Thermo Fisher Scientific). Images were collected with an inverted microscope (Nikon Eclipse TE2000-U) equipped with an illumination system X-cite 120 (EXFO Photonics Solutions Inc., Exton, PA, USA) and a cooled 12-bit CoolSnap HQ CCD camera (Photometrics, Tucson, AZ, USA). Fiji ImageJ software v1.53c was used for image processing. 

### 2.11. Statistical Analysis

Differential expression analysis was performed with NanoString nCounter Advanced Analysis software 2.0.115. Only genes with adjusted *p* < 0.1 and linear fold change ≥±1.2 were counted as differentially expressed genes (DEGs). Adjustment of *p*-values was performed with the Benjamini–Yekutieli method. For Appendix A, normalized counts for individual genes were calculated and plotted as mean ± standard deviation using GraphPad Prism 9.2.0; *p*-values for the differences between experimental samples and their corresponding controls, if significant, are shown in Appendix A. The number of individual samples in each group is provided in Table 1.

### 2.12. Data Availability

All data generated or analyzed during this study are included in this published article and its Appendix A. Mouse-adapted prion strain SSLOW used in this study is available from the lead contact but may require a completed materials transfer agreement.

## 3. Results

### 3.1. The Astrocyte Panel Detects Changes in Gene Expression Profiles upon Various Pathological Insults

For analysis of gene expression, we designed a custom NanoString panel that reports on astrocyte function genes and reactive state [28]. Gene expression was analyzed for five experimental conditions: prion infection, TBI, ischemic stroke, Aβ plaque formation, and normal aging (Table 1). For each condition, datasets were collected for four brain regions—cortex (Ctx), hippocampus (Hp), thalamus (Th), and hypothalamus (HTh) (Appendix A). 

For infecting animals with prions, we used mouse-adapted strain SSLOW. This strain has the shortest incubation time to disease among mouse-adapted prion strains, allowing us to avoid an overlap between disease-specific and age-related changes. Moreover, in SSLOW-infected mice, prions accumulate in the four brain regions studied here, causing profound neuroinflammation (Appendix A) [24,28]. Male and female C57Bl/6J mice were infected with SSLOW via intraperitoneal route (Table 1).

As a model of TBI, we employed the closed cortical impact model in adult female C57Bl/6J mice and harvested ipsilateral and contralateral brain tissues 7 and 60 days after the primary insult (Table 1), except for the hypothalamus, which was collected as a single sample containing both ipsilateral and contralateral parts. As expected for sustained injury, the gene expression profile of the ipsilateral cortex was significantly altered compared to that of the cortex in sham control animals (Appendix A). Interestingly, the ipsilateral hippocampus and thalamus displayed even more profound changes in gene expression than the cortex (Appendix A). Many genes remained upregulated 60 days post-injury, but to a lesser degree in comparison to the 7 dpi (days post-injury) group (Appendix A). In contralateral regions of the brain, only minimal gene expression changes were detectable. Therefore, for comparison with other insults, we chose the ipsilateral samples obtained 7 days after injury, which is the peak of post-traumatic neuroinflammation in this model [29,30]. 

As a model of ischemic stroke, we used MCAO in adult female C57Bl/6J mice (Table 1). The blood flow was occluded for 1 h, followed by 24 h or 5 days of re-perfusion, after which the animals were euthanized. Because brain tissue was collected from a narrow 2-mm coronal section, and due to a possible variation in the location or the extent of the stroke, MCAO samples displayed a higher degree of variability compared to the other insults (Appendix A). We chose to focus on the 24-h time point post-ischemia, which displayed a milder response but more consistent changes relative to the 5-day samples (Appendix A).

To assess changes in gene expression related to AD, we employed the 5XFAD mouse model and harvested brain tissue from 10-month-old male and female mice. These mice carry five mutations associated with a familial form of AD and at 10 months of age, exhibit behavioral deficits and a full spectrum of pathological changes, including mature amyloid plaques, profound neuroinflammation, and significant changes in gene expression (Appendix A) [21,31]. Age-matched littermates that lacked disease-associated variants (B6SJLF1/J, referred to as WT) were used for comparison (Table 1). Heat map analysis (Appendix A) showed that female 5XFAD mice had somewhat stronger changes in gene expression, which was consistent with published reports of higher plaque burden in 5XFAD females compared to males [21]. 

To analyze gene expression changes upon aging, 24-month-old C57Bl/6J male and female mice were compared with C57Bl/6J control animals of younger age: 3- and 5-month-old controls for TBI (n = 6), and normal 8 to 13-month-old mice (n = 9) (Table 1). A heat map of normalized counts for individual samples was used to visualize the level of variations between age cohorts (Appendix A). Ultimately, the youngest animals in the study, 3-month-old mice that correspond to mature adults, were used as a reference in the differential expression analysis of aged mice (24-month-old). 

### 3.2. The Region-Specificity of Astrocyte Gene Expression Is Maintained after Insults

Individual samples within the control groups demonstrated robust reproducibility and high regional specificity in their gene expression pattern (Figure 1A and Appendix A). Indeed, regardless of animal’s age, sex, or mouse strain, all control samples clustered strictly according to the brain region, showing only minor variations between different control groups (Appendix A). These data are consistent with previous reports that documented well-defined region-specific homeostatic identities of astrocytes [9,14,28,32,33]. Remarkably, in all pathological insults, the transformation of astrocytes into disease-associated states occurred in a region-specific manner, as astrocytes preserved region-specific signatures (Figure 1A and Appendix A). The preservation of regional signatures under pathological insults contrasts with the transformation of microglia, which acquire a more uniform disease-associated signature across brain regions [27]. 

### 3.3. The Degree of Astrocyte Reactivity Correlates with the Degree of Changes in Expression of Genes Associated with Astrocyte Functions

For each gene set, NanoString nCounter built-in algorithms calculated GSA scores, which reflect on a number of DEGs, weighted differences in their fold change, and statistical significance. A combined GSA score for A1-, A2-, pan-specific astrocytes, and other markers of reactive astrocytes was used as a measure of astrocyte reactivity. In a similar manner, for assessing astrocytic functional changes, combined GSA score for the other gene sets that report on astrocyte homeostatic functions was used (Figure 2).

Within individual regions, the degree of astrocyte reactivity varied depending on the insult (Figure 1B and Appendix A). Regardless of the brain region, the highest GSA scores, which reported on astrocyte reactivity and functional changes, were found in prion-infected animals (Figure 1B and Appendix A). Plotting a combined undirected GSA score that reflects astrocyte reactivity against the combined undirected GSA score for physiological functions revealed a very strong correlation between the two (Figure 2). Remarkably, as judged from the R^2^ values, the strength of the relationship was very high in all four regions examined (Figure 2). These results suggest that, regardless of the nature of the insult or the identity of astrocyte reactive phenotype, the changes in functional pathways are tightly linked with astrocyte reactivity.

### 3.4. Under Pathological Insults, Astrocytes Exhibit Region-Specific Continuums of Gene Expression Signatures

To analyze region-specific insult-elicited differences across all groups, we performed a PCA that placed the individual samples of different insults into a continuum of gene expression signatures (Figure 3). The PCA of the entire panel resulted in two well-resolved continuums: one shared by the cortex and hippocampus, and another shared by the thalamus and hypothalamus (Figure 3A, left panels). Within each continuum, we observed overlaps between individual insult-dictated gene expression signatures (Figure 3A, right panels). As expected, the gene expression signatures in normal controls overlapped substantially with those of the Aging group. Furthermore, Aging overlapped with 5XFAD and MCAO-24h, 5XFAD and MCAO-24h overlapped with TBI-7d, and finally, TBI-7d overlapped with prions. We observed similar patterns for smaller gene sets that report on specific functions, including neuroprotection/neurotoxicity, transporters, metabolism, reactive astrocytes, and others (Figure 3B). However, depending on the gene set employed, a better separation of cortex- versus hippocampus-specific continuums (metabolism) or thalamus- versus hypothalamus-specific continuums (transporters, neuroprotection/neurotoxicity) can be seen (Figure 3B). 

To summarize, (i) astrocytic gene expression signatures associated with different pathological insults partially overlapped and, together, constituted continuums; (ii) individual continuums were region-specific. In contrast to astrocytes, the microglia gene set, which in PCA consisted of 13 genes (Il1a, Tnf, Tmem119, Rgs10, Socs3, Gas1, P2ry12, Cd68, Tlr2, Ccl2, Csf1, as well as DAM signature genes Apoe and Vegfa) did not form well-defined region-specified phenotypes, showing mostly insult-specific differences, which were partially overlapping (Figure 3B). For neurons, a set of 10 genes (Arc, Gabrg1, Grin1, Grin2b, Grm2, Nos1, Slc32a1, Snap25, Syn2, Syp) was able to detect a well-defined separation according to brain region but showed only a weak insult-specific separation within individual regions (Figure 3B).

### 3.5. Insult-Specific Changes in Astrocyte Gene Expression Signatures

A significant overlap in DEGs between experimental conditions was detected (Figure 4A, Appendix A), bringing up the possibility that the same genes are involved in remodeling regardless of insult. Of particular interest is the Aged group, which shared a significant number of DEGs with each studied insult (Figure 4B; Appendix A). Out of 32 DEGs identified in the cortex of the Aged cohort, the number of DEGs shared with other conditions increased from 53% for 5XFAD and 56% for MCAO-24h to 81% for the TBI-7d and prion groups (Figure 4B). On top of the upregulated DEGs that were common across all conditions including Aging were markers of reactive astrocytes GFAP, Vim and Serpina3n, complement factor C4a, and the gene related to astrocyte function Slc14a1 (urea transporter) (Appendix A). 

The strong correlation between astrocyte reactivity and the extent of the changes in function genes (Figure 2) concealed important insult-specific nuances that became evident upon differential expression analysis. The number of detected DEGs in the cortex gradually increased from 32 in the Aged group and 50 in the 5XFAD group, to 139 in MCAO-24h, 151 in TBI-7d, and 226 in Prions (Appendix A). Venn diagrams of cortical DEGs built separately for the markers of reactive states and function-related genes showed that the functional DEGs had greater insult-specificity than the reactive marker gene set (Figure 4C). Remarkably, this trend was observed between groups with strong changes in gene expression (Prions, TBI-7d, MCAO-24h), as well as groups with mild changes (5XFAD and Aged mice) (Figure 4C). Although the degree of astrocyte reactivity correlates with the extent of dysregulation in genes associated with astrocyte function, the functional changes were not identical between the insults. 

Linear fold changes in genes related to homeostatic functions were much lower than the changes in the markers of reactive states (Appendix A). Clustering the log2 linear fold changes for all experimental groups revealed that both sets of genes, i.e., those reporting on astrocyte reactivity and functional dysregulation, displayed insult-specific response patterns (Figure 5A,B). For instance, markers of reactive astrocytes GFAP and Serpina3n clustered together, while Vim belonged to a different cluster, pointing to the existence of disease-specific response patterns of astrocyte reactivity (Figure 5A). The same was true for the set of genes associated with astrocyte functions (Figure 5B). For example, the BBB regulator Agt was detected among DEGs common for Prions and TBI-7d, but it changed in the opposite directions: upregulated in TBI-7d, but downregulated in Prions (Appendix A). MCAO-24h was characterized by a transient downregulation of many genes associated with astrocyte functions (Figure 5B) that were upregulated in TBI-7d. The expression of the most prevalent water channel, Aqp4, increased in the cortices of 5XFAD, TBI-7d, and Prions, but remained unchanged in MCAO-24h (Appendix A). Channels and transporters were affected in Prions, TBI-7d, and MCAO-24h, but out of these three animal groups, the downregulation of Kcna2 and Kcnk1 was accompanied by a statistically significant upregulation of ATPase subunits in the Prions group only (Appendix A). Nevertheless, despite the insult-specificity in astrocyte response, the net changes in functional gene sets correlated well with the net changes in astrocyte reactivity (Figure 2). 

### 3.6. Region-Specific Changes in Astrocytic Phenotypes

To study the extent of astrocyte transformations in different brain regions, we focused on cortex, hippocampus, thalamus, and hypothalamus in the Prion group. Prions induced the strongest gene expression changes throughout the brain, while other insults preferentially affected certain regions and/or had a milder impact (Figure 1B and Appendix A). Despite maintaining region-specificity of astrocytes even in the Prion group (Figure 1A), a remarkable overlap of DEGs between Ctx, Hp, Th, and HTh was observed (Figure 6A). These findings suggest that, similar to the insult-specific changes in astrocytic phenotypes, regional transformation involves quantitative changes of common genes, regardless of the brain region. 

### 3.7. Thalamus Is Vulnerable, Regardless of the Nature of the Pathological Insult

In animals challenged with the prion strain SSLOW, prion replication is widespread across the brain [24,28], whereas other experimental insults used here were expected to target brain regions more selectively. For instance, in TBI and MCAO, the primary impact of injuries was expected to affect the cortex due to the primary injury location and occlusion region. In 5XFAD mice, the earliest and heaviest plaque deposition was found in the cortex and hippocampus [21]. Surprisingly, in the current work, the thalamus showed profound astrocyte response in all experimental insults (Figure 6B–I). As judged from the GSA scores, the high vulnerability of the thalamus was especially noticeable in TBI, despite the fact that the cortex was the site of injury (Figure 6B). Moreover, in TBI, the number of DEGs in the ipsilateral thalamus exceeded those in the contralateral cortex 7 days post insult and remained high, even at 60 days post TBI (Figure 6D,E). For comparison, HTh showed only a minimal number of DEGs in TBI-7d (Figure 6D). In the Prion group and in 5XFAD mice, the number of DEGs in the thalamus was higher than in other regions (Figure 6C,H). In 5XFAD mice, the number of DEGs in Th exceeded the number of DEGs in Ctx and Hp, while HTh was minimally affected (Figure 6H). In comparison to animal groups subjected to insults, the natural Aged group showed a mild response of Th with respect to the number of DEGs (Figure 6I), but a relatively strong response with respect to the GSA score (Figure 6B). In summary, the thalamus appears to be a highly vulnerable brain region, reacting not only to a direct insult but also to injuries delivered to other brain regions.

## 4. Discussion

Studies on single-cell transcriptional profiling revealed that, under normal conditions, there are seven distinct, developmentally predetermined sub-types of astrocytes with well-defined regional specialization in the mouse brain [9,10]. Physiological and morphological studies demonstrated that astrocytes are indeed functionally diverse and form distinct neural circuit-specialized subpopulations in different brain regions [6,7,8,9,10,11,12,13]. However, considerably less is known about the phenotypic diversity of astrocytes under pathological conditions [14,15,16,17]. In recent years, several important questions have been under critical discussion. Do astrocytes respond to pathological insults in a specific manner by adopting distinct, insult-specific states? Are insult-specific states uniform across brain regions? What are the roles of insult-specific response versus region-specific homeostatic identity in dictating astrocyte’s reactive phenotype [34]? 

To answer these questions, current studies have examined the region-specific response of genes associated with astrocyte functions and reactivity to insults of diverse natures, including prion infection, mechanical injury (TBI), genetic mutations associated with familial AD, ischemic insult, and normal aging. We found that, under pathological insults, expression of genes associated predominantly with astrocytes preserved region-specific signatures, suggesting that astrocytes respond to insults in a region-specific manner. Comparing the gene sets that report on homeostatic functions with those that report on astrocyte reactivity, we found that the former displayed a higher level of disease-specificity relative to the latter. In fact, common genes that were found to be involved in astrocyte remodeling across insults and normal aging consisted of markers of astrocyte reactivity such as GFAP, Vim, and Serpina3n. Remarkably, regardless of the nature of an insult or the insult-specificity of astrocyte response, strong correlations between the degree of astrocyte reactivity and perturbations in their homeostasis-associated genes were observed within each individual brain region. These results suggest that astrocyte reactivity changes in parallel with changes in the expression of genes associated with astrocyte functions, regardless of the nature of an insult. Because our study does not analyze transcriptome under single-cell resolution, the current results do not dismiss the idea regarding the existence of insult-specific phenotypes of reactive astrocytes. The astrocyte gene-targeted approach did not separate the insult-specific gene expression signatures well from each other, which instead partially overlapped, forming continuums. Surprisingly, the continuums were region-specific, highlighting the role of region-specific homeostatic identity in defining astrocytic response to insults. Consistent with the last statement, thalamic astrocytes were found to be the most responsive to insults, as they reacted not only to direct insults within the thalamus, but also to injuries delivered to other brain regions.

How many reactive phenotypes of astrocytes exist? The abandonment of the A1 (neurotoxic)/A2 (neuroprotective) binary model has renewed the discussion about the diversity of reactive phenotypes [28,34,35,36]. Previous studies of individual neurodegenerative diseases, including prion disease and multiple sclerosis, as well as peripherally induced neuroinflammation, have documented that astrocytes respond to insults in a region-specific manner and maintain their region-specific identities [17,28,37,38,39]. Several studies have focused on the diversity of astrocytes within individual diseases [14,17,35,40], while a region-specific profiling of reactive phenotype across multiple diseases has never been performed. Is it the nature of an insult or region-specific homeostatic identity that drives the reactive phenotype? On the one hand, changes in the expression of homeostatic genes within the same brain region showed limited overlaps between individual insults, suggesting that the nature of an insult plays an important role in dictating the range of homeostatic functions affected (Figure 4). On the other hand, PCA, which considers not only the number of changed genes but also the extent of their changes, revealed a better separation into distinct clusters based on brain region rather than the nature of an insult. For instance, even within the same insult, the gene expression signatures of cortical and thalami clusters were clearly distinctive (Figure 3A). Depending on the specific homeostatic gene set, better separations between the region-specific signatures of the cortex and hippocampus or thalamus and hypothalamus could be achieved (Figure 3B). Moreover, within the same region, the gene expression signatures for individual insults showed considerable overlaps. Together, these results suggest that region-specific homeostatic identities are important in specifying astrocyte response to pathological insults. 

In agreement with the result of the current work, our previous study unveiled significant, region-specific differences in the astrocytic response to prion infection. This was assessed through an examination of the cell morphology, subcellular localization, and gene expression patterns of GFAP, S100b, and Aldh1l1, utilizing immunohistochemistry and qRT-PCR techniques, respectively [37]. In contrast to the thalamus, where the changes in astrocyte morphology appeared subtle in response to prions, reactive astrocytes within the cortex and hippocampus exhibited significantly enlarged cell bodies and thickened processes. The region-specific differences in the morphology of reactive astrocytes and their immunofluorescence patterns align harmoniously with the PCA results presented in Figure 3, which also document a close proximity between cortical and hippocampal continuums, but clear separation between the continuum of the thalamus and those of the cortex and hippocampus. 

Analysis of the affected physiological pathways suggests that in the reactive phenotypes, a global transformation of the homeostatic functions took place, as evident from the disturbance in multiple pathways including BBB regulation, transporters, neuroprotection and neurotoxicity, extracellular matrix, myelination, lipid metabolism, and others (Appendix A). Remarkably, the GSA score that reflect dysregulation in homeostatic functions correlated strongly with the GSA score that reports on astrocyte reactivity. Are these data purely correlative, or does a causal relationship between astrocyte reactivity and homeostatic function exist? Within individual brain regions, the relationship between astrocyte reactivity and change in homeostatic functions was strong across insults of different natures and was not affected by the degree of impact on individual regions. Moreover, regardless of a specific reactive phenotype or function affected by individual insults, the net changes in homeostatic gene sets mirrored the degree of astrocyte reactivity, suggesting that the two parameters are tightly coupled and reflective of a global transformation in astrocyte physiology. Whether the net impact of such transformation is beneficial or detrimental is difficult to project from transcriptome analysis alone and requires functional tests. In previous studies, reactive astrocytes isolated from prion-infected animals or mouse models of AD displayed a reduced expression of neuronal support genes and was found to be synaptotoxic in neuronal-astrocyte co-cultures [41,42]. 

The observed correlation between gene scores reflecting changes in homeostatic functions and astrocyte reactivity could be attributed to proliferation of reactive astrocytes, causing all astrocyte-associated genes to change proportionally with the level of proliferation. Reactive astrocytes do undergo proliferation in response to certain insults, such as scar formation [18]; however, this phenomenon is not witnessed across all pathological insults. If this hypothesis is correct, one would expect the highest astrocytic proliferation in the prion group, since prion-infected mice showed the highest GSA scores. However, the lack of astrocyte proliferation in prion disease has been well established. Cell-type-specific ribosomal profiling and single-cell transcriptome analyses revealed that in prion-infected mice, the number of astrocyte do not change or might even decline with the disease progression [43,44]. Examination of plaque-forming AD mouse models using cell proliferation assays revealed the proliferation of microglia, but a lack of such for astrocytes [45]. As for mice subjected to MCAO, it is difficult to anticipate significant astrocyte proliferation within a 24-h timeframe following the insult. In cases of TBI, astrocyte proliferation does occur, but primarily at the site of the primary injury [46]. Notably, even though the cortex served as the principal site of TBI injury, in the current work the thalamus displayed the highest GSA scores in relation to both astrocyte reactivity and function within the TBI group. Were the correlation solely attributable to astrocyte proliferation—in other words, the biological size—one would anticipate such a connection exclusively within the primary injury sites, namely the cortex. Intriguingly, the correlation was observable across all four brain regions, irrespective of the expected astrocyte proliferation in the primary lesion sites. This observation suggests that proliferation is not the predominant factor underpinning this correlation.

Although mice in the Aged group were not provoked by an insult, consistent with previous studies [47,48], their astrocytes displayed clear signs of transformation into reactive phenotypes. Two interesting observations can be made. First, the genes that were dysregulated across all groups predominantly consisted of genes that report on astrocyte reactivity, with only one gene associated with homeostatic functions. Second, the percentage of overlap between DEGs in the Aged group and those in other groups increased with the severity of astrocytic response to individual insults, from 53% overlap with the 5XFAD group to 81% overlap with the Prion group (Figure 4B). Although the clusters corresponding to the Prion and the Aged groups were separated far apart in the PCA, primarily due to differences in severity of the responses (Figure 3), the DEGs of the Aged group overlapped almost entirely with the DEGs of the Prion group. This suggests that astrocytic response to prions and other insults engage and perhaps overuse the pathways that have evolved for optimizing astrocyte homeostatic function in normal aging. 

The lower scores in the 5XFAD model relative to the prion-infected mice could be attributed to several factors. First, the 5XFAD mice might exhibit inherently milder disease pathology compared to prion-infected mice. Indeed, after developing a full spectrum of pathological changes by 6–7 months of age, 5XFADs endure pathology well for at least a year, showing only very mild disease progression [31]. Second, to avoid an overlap between disease-specific and age-related changes, we analyzed 10-month-old 5XFAD, a stage of the disease that is likely to be less advanced relative to that of prion-infected mice. Finally, in 5XFAD mice, astrocytes are activated only in close proximity to Aβ plaques and, in bulk tissue analysis, are diluted with homeostatic astrocytes. 

Are region-specific populations of astrocytes equally vulnerable to insults? As judged by changes in the transcriptome, the rates of astrocytes aging under normal conditions varied for different brain regions [48]. The vulnerability of thalamic astrocytes in prion diseases could be attributed to the intrinsic tropism of prions to this region, since the thalamus is impaired at the early stages and is the most severely affected at the advanced stages of prion diseases [27,49,50]. In 5XFAD mice, the cortex and hippocampus are the first regions to deposit Aβ plaques and show signs of reactive astrocytes at younger ages [21]. However, by 14 months of age, the thalamus exhibited the highest load of Aβ plaques in 5XFAD mice as judged by the uptake of an amyloid-specific tracer [51]. In the current work, thalamic astrocytes reacted not only to the insults known to target the thalamus, but also to injuries delivered to other brain regions. In fact, among the four regions, the thalamus showed the highest GSA scores with respect to both astrocyte reactivity and function in the TBI group (Figure 6B). These results are consistent with previous clinical work that employed a positron emission tomography ligand for measuring glia activation in humans. Up to 17 years after severe TBI, individuals showed profound chronic neuroinflammation in the thalamus, which was attributed to damages in the thalamo-cortical tract [52,53]. Moreover, neuroinflammation in the thalamus has been proposed to serve as a marker of cortical injury and subsequent long-term cognitive deficits [54]. In the Aged group, the thalamus showed the highest GSA scores for the functional gene sets (Figure 6B). As the thalamus receives reciprocal projections from the entire cerebral cortex, its vulnerability could play a significant role in understanding pathology and predicting disease outcomes [53,55].

Several intrinsic limitations of the current work are worth discussing. First, examining the transcriptome of bulk tissues does not report on the actual heterogeneity or identity in cell reactive phenotypes within individual brain regions, or the diversity of cell responses to individual insults. Second, while the panel consists of genes that were selected to report on astrocyte function or reactive state, other types of brain cells likely make some contributions to the average of gene expression measured, which likely differs based on brain region. The contributions of other cell types cannot be excluded, especially under pathological conditions. Third, the analysis of transcriptome on its own is not sufficient for defining functional phenotypes of the reactive state or making conclusions regarding the net neurotoxic or neuroprotective impact of astrocyte transformation. Moreover, region-specific differences in the severity of astrocyte response can be interpreted in several alternative ways—for instance, as differences in intrinsic vulnerability of astrocytes to insults, or as differences in the strength of physiological response, whether maladaptive or not, to restore homeostasis. The current data should not be used for tracking the dynamics of individual genes, but rather for profiling of region-specific response. Sex-specific differences were previously found to influence glia reactivity in neurodegeneration associated with TBI and Alzheimer’s disease [29,56]; however, they were not the subject of the current work. Considering the results and limitations mentioned above, the current work supports the idea that rigorous analyses of astrocyte transcriptome on a single-cell level requires knowledge about the regional identity of astrocytes, especially for comparing the data acquired by independent studies.

Despite limitations, targeted approaches that involve bulk tissue analyses might be well-positioned for comparing a large number of samples across various pathological conditions. Its value is in providing a holistic view, complementary to single-cell RNAseq approaches for monitoring a response to pathological insults. Astrocytes translate mRNAs in their peripheral processes that are adjacent to synapses and enriched with mRNAs associated with biological functions, including regulators of synaptic plasticity, GABA, glutamate metabolism, and lipid synthesis [57]. In a mouse brain, one astrocyte is estimated to contact up to 100,000 synapses, whereas a human astrocyte can contact up to 2,000,000 synapses [58]. Due to significant losses of peripheral astrocyte processes during isolation, the approaches that rely on isolation of single cells or nuclei are likely to misrepresent changes in gene sets that report on astrocyte functions. Another potential problem associated with the single-cell RNAseq approach is assignment of cell types, which is dependent on cell identity markers. If during pathological insults a cell changes the expression of the cell identity markers used for identification, that cell becomes wrongly assigned to be a different cell type.

Recent work that employed single-nucleus RNAseq revealed a subpopulation of astrocytes in aged wild type mice and human brain that was phenotypically similar to a subpopulation of disease-associated astrocytes identified in the 5XFAD AD mouse model [14]. While the approach in the current study lacked resolution of a single-nucleus RNAseq, we also observed considerable overlap in clusters corresponding to the same mouse model of AD and 24-month-old aged wild-type mice. Remarkably, our study arrived at similar conclusions despite intrinsic limitations, such as a limited resolution due to analysis of only certain genes in bulk tissue. Nevertheless, the current approach could be useful for region-specific profiling across multiple insults in humans. 

## Figures and Tables

**Figure 1 cells-12-02172-f001:**
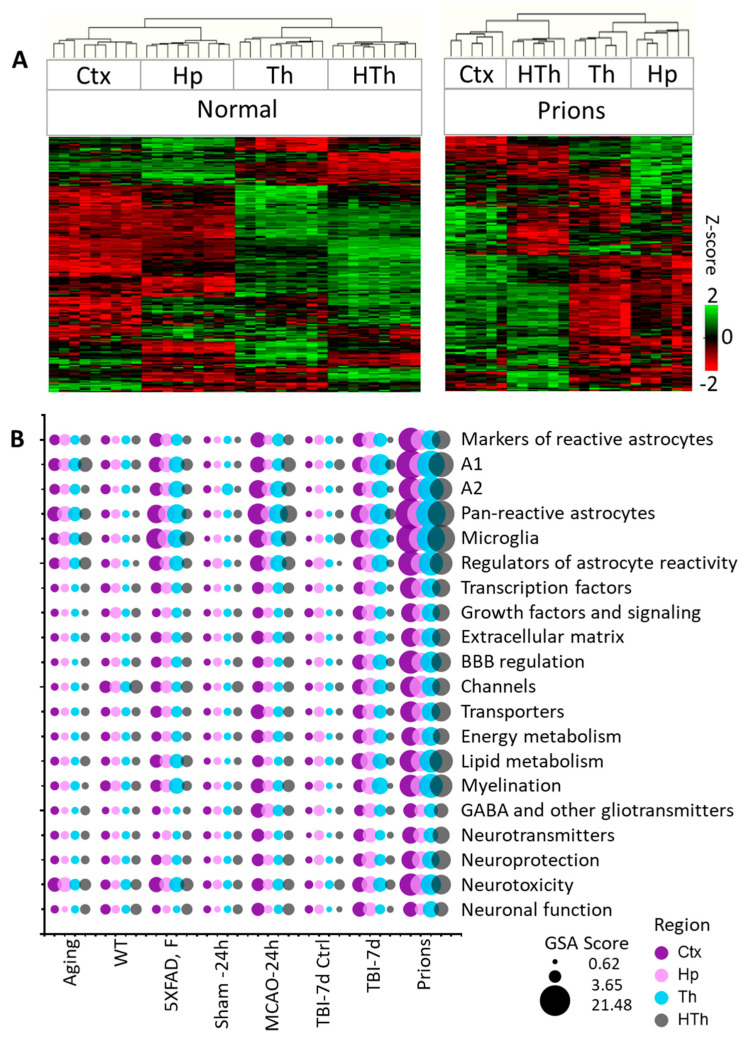
Analysis of gene expression in different brain regions. (**A**) Hierarchical clustering of mock-inoculated samples from the Normal group (left) and Prions group (right) showing high reproducibility of the astrocyte panel and regional specificity of astrocyte function genes, which is maintained at the terminal stage of prion infection. Clustering was performed using genes related to BBB regulation, lipid and energy metabolism, extracellular matrix, junction, myelination, channels, transporters, gliotransmitters and neurotransmitters, neuroprotection and neurotoxicity, and astrocyte-specific markers (Appendix A). (**B**) Bubble plot of undirected global significance scores obtained with NanoString Gene Set Analysis (GSA) by comparing all experimental and corresponding control groups, with the Normal group used as a reference (Table 1). TBI-7d is represented by ipsilateral Ctx, Hp and Th, and whole HTh. The size of the bubbles reflects the degree of regional changes for individual gene sets with different insults. Ctx—cortex, Hp—hippocampus, Th—thalamus, HTh—hypothalamus.

**Figure 2 cells-12-02172-f002:**
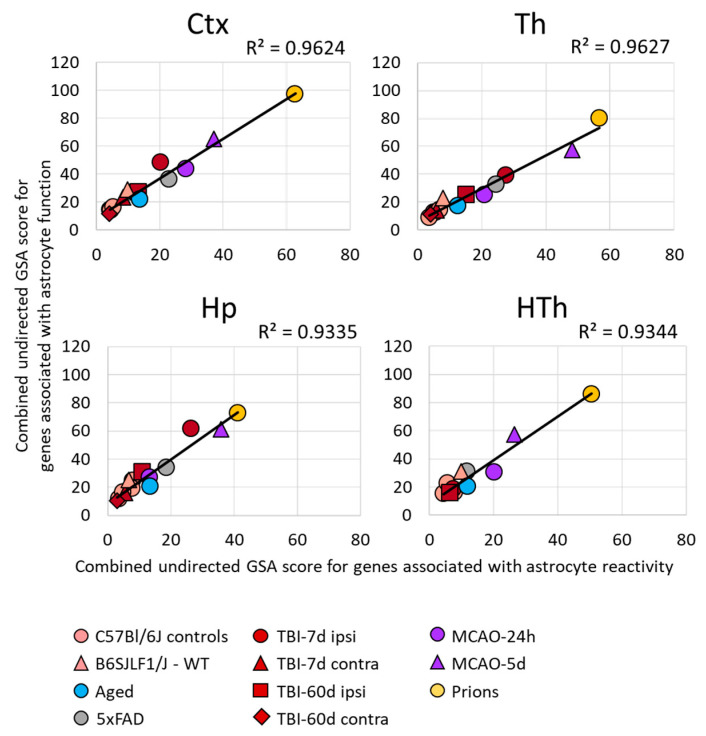
Relationship between the changes in the expression of gene sets in different brain regions. A strong correlation between the sum of undirected GSA scores of the gene sets reporting astrocyte reactivity (A1-, A2-, pan-specific markers, and other markers of reactive astrocytes) and dysregulation of astrocyte function (BBB regulation, channels, lipid metabolism, myelination, energy metabolism, extracellular matrix, gliotransmitters, neurotransmitters, transporters, neuroprotection, neurotoxicity). Each symbol represents a distinct experimental or control group, as indicated. Ctx—cortex, Hp—hippocampus, Th—thalamus, HTh—hypothalamus.

**Figure 3 cells-12-02172-f003:**
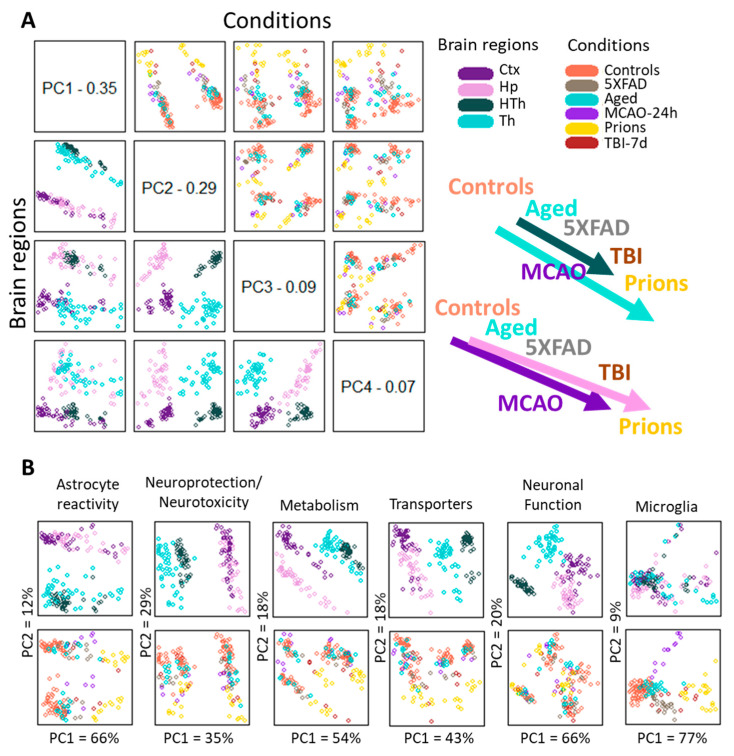
Principle Component Analysis. (**A**) PCA performed using the entire panel of genes for samples from five experimental groups and four brain regions. The control group in the PCA included Normal, TBI-7d Ctrl, and Sham-24h samples combined. The distribution of samples according to brain region and condition is shown on the left bottom and top right plots, respectively. A schematic illustration of continuums of astrocyte gene expression signatures is shown on the right. (**B**) PCA performed using selected gene sets for the same samples. In panels (**A**,**B**), each dot represents an individual animal, with different colors representing different brain regions or experimental conditions. Ctx—cortex, Hp—hippocampus, Th—thalamus, HTh—hypothalamus.

**Figure 4 cells-12-02172-f004:**
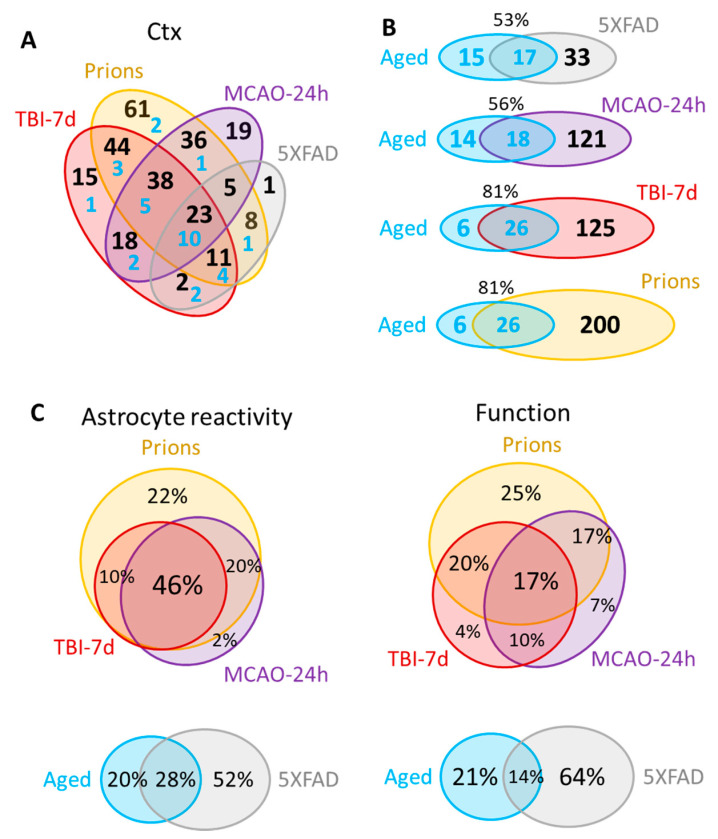
Venn diagrams for the differentially expressed genes in the cortex. (**A**) Venn diagram for the DEGs in the cortex of 5XFAD (females and males, n = 3 + 3), MCAO-24h (females, n = 3), TBI-7d (females, n = 3), and Prions (females and males, n = 3 + 3). Blue numbers show the portion of DEGs that are common with the Aged group (females and males, n = 3 + 3). (**B**) Venn diagrams representing overlaps in DEGs detected for Aged cortex and for cortices of other groups. (**C**) Venn diagrams of the DEG percentage overlap in the reporters of astrocyte reactivity (**left**) and function (**right**), presented for the groups with strong changes in gene expression (**top**) and mild changes in gene expression (**bottom**). Genes annotated as A1, A2, pan-reactive, other markers of reactive astrocytes, regulators of astrocyte reactivity, and astrocyte-specific markers were considered as reporters of reactivity. Genes related to BBB regulation, lipid and energy metabolism, extracellular matrix, junction, myelination, channels, transporters, gliotransmitters and neurotransmitters, and neuroprotection were included into Function DEGs. Differential expression analysis was performed using NanoString Advanced Analysis software 2.0.115. Each group was compared to their corresponding controls (Table 1). DEGs for the Aged group were calculated relative to the youngest control group (TBI-7d Ctrl, n = 3F). The genes with linear fold greater than 1.2 up or down and adjusted *p* < 0.1 were selected as DEGs (Appendix A). Ctx refers to the cortex. TBI-7d is represented by the ipsilateral Ctx.

**Figure 5 cells-12-02172-f005:**
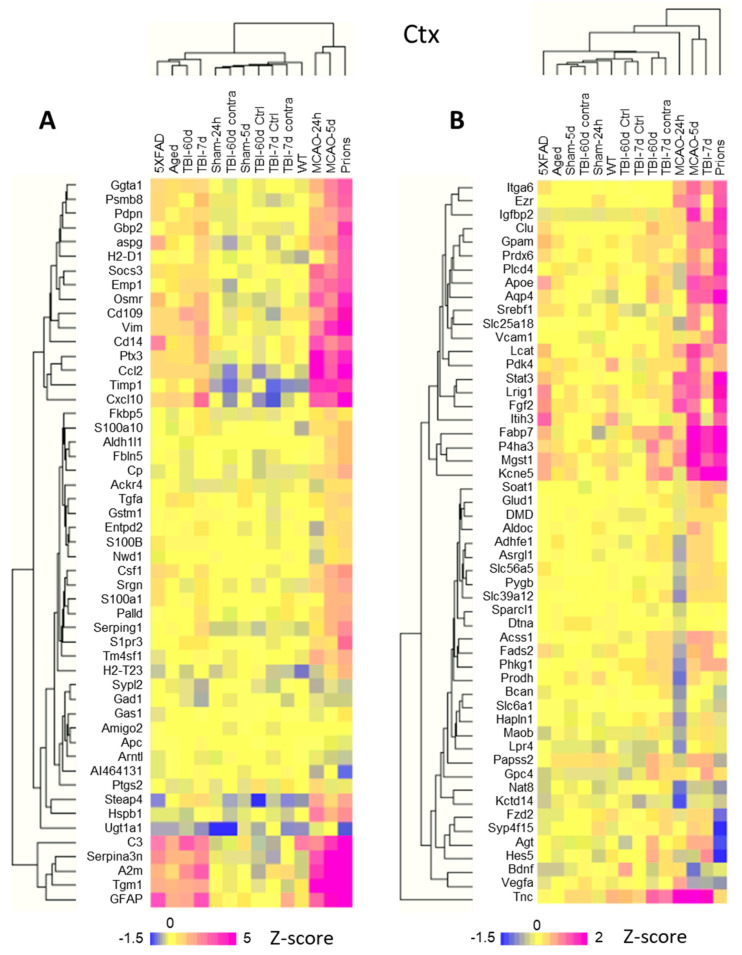
Hierarchical clustering of log2 changes in the expression of selected genes in the cortex. (**A**) Selected astrocyte reactivity reporters form several separate clusters displaying an insult-specific pattern of activation. (**B**) Hierarchical clustering of selected astrocyte function genes highlights insult-specific changes in their expression. Hierarchical clustering was performed for groups with a number of samples, as described in Appendix A. TBI-7d and TBI-60d represent ipsilateral samples. Z-score transformation was applied to the genes.

**Figure 6 cells-12-02172-f006:**
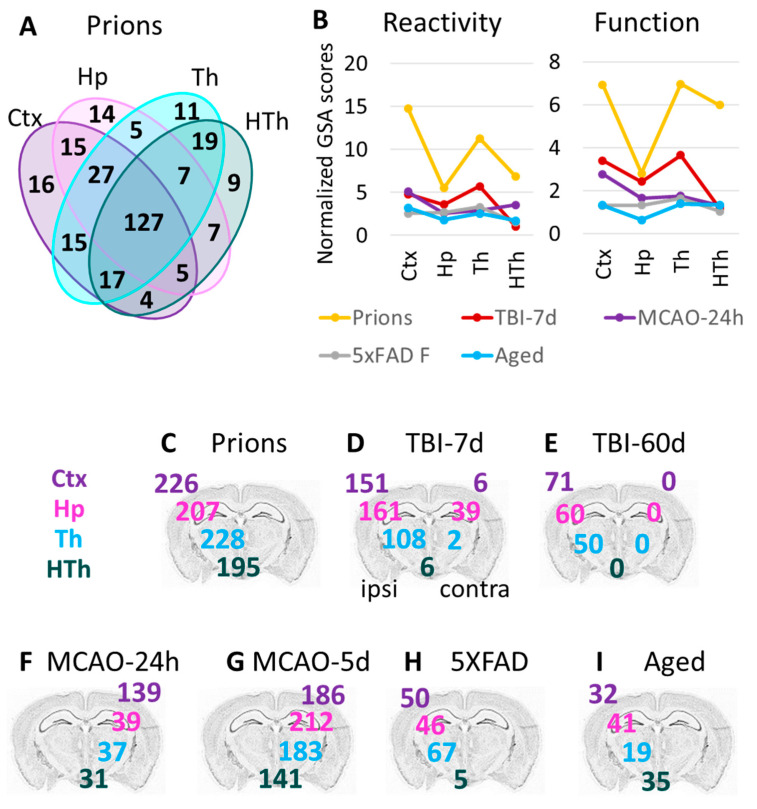
Region-specific changes in gene expression. (**A**) Venn diagram for DEGs in four brain regions of the Prions group (females and males, n = 3 + 3) compared to the Normal group (females and males, n = 6 + 3). (**B**) Combined undirected global significance scores characterizing astrocyte reactivity (**left**) and function (**right**) in four brain regions after five insults. Undirected global significance scores were calculated with gene set analysis (GSA) for individual gene sets and were then summed to obtain combined GSA scores for Reactivity and Function for each brain region and insult. The combined GSA scores for Reactivity included GSA scores for A1-, A2-, pan-specific markers, and other markers of reactive astrocytes. The combined GSA scores for Function included GSA scores for BBB regulation, lipid and energy metabolism, extracellular matrix, myelination, channels, transporters, gliotransmitters and neurotransmitters, and neuroprotection. To normalize, the combined GSA scores for each insult were divided by the corresponding GSA scores obtained for the Normal group. TBI-7d is represented by ipsilateral Ctx, Hp and Th, and whole HTh. (**C**–**I**) The number of differentially expressed genes (DEGs) in all experimental groups (Table 1): Prion (**C**), TBI-7d (**D**), TBI-60d (**E**), MCAO-24h (**F**), MCAO-5d (**G**), 5XFAD (**H**), and Aged (**I**). The numbers for DEGs in the cortex, hippocampus, thalamus, and hypothalamus are displayed over the corresponding regions of the brain. Differential expression analysis was performed using NanoString Advanced Analysis software 2.0.115. Each group was compared to its corresponding controls (Table 1). DEGs for the Aged group were calculated relative to the youngest control group (TBI-7d Ctrl, n = 3F).

**Table 1 cells-12-02172-t001:** List of animal groups analyzed using Astrocyte panel.

Group Name	Mouse Strain	n	Age at Insult	Insult	Time of Euthanasia
Prions	C57Bl/6J	4F + 2M ^1^	6 weeks	1% SSLOW i.p.	Terminal stage, dpi: F—166, 166, 173, 173 ^2^ M—175, 185
Normal (Norm)	C57Bl/6J	6F + 3M	6 weeks	1× PBS i.p.	dpi: F—197, 223, 223, 295, 346, 363; M—203, 225, 229
TBI-7d ipsilateral	C57Bl/6J	3F	12 weeks	Left hemi CCI ^3^	7 dpi (3 months old)
TBI-7d contralateral					
TBI-60d ipsilateral	C57Bl/6J	3F	12 weeks	Left hemi CCI	60 dpi (5 months old)
TBI-60d contralateral					
TBI-7d Ctrl	C57Bl/6J	3F	n/a	Sham/no CCI	3 months old
TBI-60d Ctrl	C57Bl/6J	3F	n/a	Sham/no CCI	5 months old
MCAO-24h	C57Bl/6J	3F	5 month	60 min MCAO ^4^	1 dpi
MCAO-5d	C57Bl/6J	3F	5 month	60 min MCAO	5 dpi
Sham-24h	C57Bl/6J	3F	5 month	Sham	1 dpi
Sham-5d	C57Bl/6J	3F	5 month	Sham	5 dpi
Aged	C57Bl/6J	3F + 3M	n/a	Aging	24 months old
5XFAD	B6SJL	3F + 3M	n/a	5XFAD ^5^ mutations	10 months old
WT	B6SJL	3F + 3M	n/a	No mutations	10 months old

^1^ F—females, M-males; ^2^ dpi—days past insult; ^3^ CCI—Controlled Cortical Impact; ^4^ MCAO—Middle Cerebral Artery Occlusion; ^5^ FAD—Familiar Alzheimer’s Disease.

## Data Availability

All data generated or analyzed during this study are included in this published article and its Appendix A.

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
