# Peer review of "Region-Specific Homeostatic Identity of Astrocytes Is Essential for Defining Their Response to Pathological Insults"

_cells, 2023, doi:10.3390/cells12172172_

Round 1

Reviewer 1 Report

The manuscript by Natallia Makarova and colleagues used transcriptomics to define region-specific changes of reactive astrocytes in five different pathological situations. The authors addressed an important and topical question, and they describe several significant outcomes. Overall finding is that astrocytes acquire reactive phenotypes depending on region-specific homeostatic identities, with an impressive overlap between the individual insults. Very interesting is also how the four pathological insults relate to the aging model.

There are a few minor issues the authors should address. A rationale for analyzing the four brain regions should be provided, including why cerebellum was not included. Authors should also improve on the rationale for using this synthetic prion strain (SSLOW), which causes widespread neuroinflammation rather than targeted inflammation of disease-specific regions. It is unclear why this model was chosen in a study focused on region-specific astrocyte activity and homeostasis. Regarding the 5XFAD AD model, the methodology is absent from the Methods section. The experimental endpoint of these mice is not described. The description of control (WT) animals should be in the Methods section also. Regarding the MCAO model of insult, the description of control animals is lacking. Table S2 could be included as a main table in the manuscript. Much of the content in the Results section should be placed in the Methods section. There is some redundancy throughout the manuscript. Examples include repeated defining of acronyms and repeated defining of conditions for counting genes. In Introduction and Discussion, 7 subtypes of astrocytes are mentioned, but no details are provided, or how these subtypes were modulated in the 5 pathological situations studied. These are all very points, and easy to address.

Taken together, the authors provide important data which are novel and significant. The manuscript is well done, experiments are overall clearly described and well controlled, and conclusions are justified by the experimental data.

Minor points:

The authors use for the plural of continuum continuums, not clear whether continua is more appropriate. Please correct the number in the title of Table S2.

Please correct the 5XFAD label in figure 4C comparing astrocyte function.

Please correct the typo in Nanostring in the caption of Figure 1B.

Author Response

We sincerely appreciate the reviewers for their time and effort dedicated to reviewing our manuscript. We thank the reviewers for acknowledging the significance of our research and for their constructive feedback that has been invaluable for improving the quality and clarity of our research. All changes in the revised manuscript are tracked.

Comment: A rationale for analyzing the four brain regions should be provided, including why cerebellum was not included.

Response: This is excellent question. We limited our analysis to those four brain regions, that we considered to be affected the most across pathological conditions studies here. Cerebellum was not included, because it was not expected to be affected in TBI or MCAO.

Comment: Authors should also improve on the rationale for using this synthetic prion strain (SSLOW), which causes widespread neuroinflammation rather than targeted inflammation of disease-specific regions. It is unclear why this model was chosen in a study focused on region-specific astrocyte activity and homeostasis.

Response: Thank you for allowing us to clarify the choice of SSLOW. In prion diseases, brain regions affected by the disease are determined by tropism of prion strains (in mice) or disease subtype (in humans) to different brain areas. In comparison to other prion strains, SSLOW is less selective with respect to brain region it affects, therefore SSLOW-associated neuroinflammation is more widespread relative to other prion strains. Such characteristic permit comparison of prion diseases with other insults.  In addition, among mouse-adapted strains, SSLOW has the shortest incubation time to disease. We thought that SSLOW will allow us to look at truly prion-specific changes avoiding changes due to normal aging.  We revised the sentence that provide the rationale for using SSLOW.

Comment: Regarding the 5XFAD AD model, the methodology is absent from the Methods section. The experimental endpoint of these mice is not described. The description of control (WT) animals should be in the Methods section also.

Response: In the revised manuscript, description of 5XFAD mice and corresponding controls is included in Methods. All 5XFAD and control mice were euthanized upon reaching 10-month of age (indicated in the Table 1 and the main text), when they exhibit a full spectrum of pathological changes, including mature amyloid plaques. 

Comment:  Regarding the MCAO model of insult, the description of control animals is lacking.

Response: Revised methods now include description of the control group.

Comment:  Table S2 could be included as a main table in the manuscript. Much of the content in the Results section should be placed in the Methods section.  There is some redundancy throughout the manuscript. Examples include repeated defining of acronyms and repeated defining of conditions for counting genes. Response: Following this recommendation, Table S2 is moved to the main manuscript (now Table 1). To avoid redundancy, we removed description of Nanostring gene panel from the Result section. We also double-checked and removed redundancy in defining acronyms. 

Comment:   In Introduction and Discussion, 7 subtypes of astrocytes are mentioned, but no details are provided, or how these subtypes were modulated in the 5 pathological situations studied. These are all very points, and easy to address.  Response: 7 subtypes of astrocytes (Zeisel et all Cell 2018) were identified using single-cell transcriptional profiling. In the current work, we analyzed expression of astrocyte-specific genes in bulk tissues of four brain region. Our approach provide a ‘bird-view’, but lacks the resolution necessary to track subpopulations identified by single-cell approach. For this reason, discussing how phenotypes of these subpopulation change would be an overreach.

Minor points:

The authors use for the plural of continuum continuums, not clear whether continua is more appropriate. Response: per Britannica Dictionary, both continuums and continua could be used as the plural of continuum.

Please correct the number in the title of Table S2.  Response: Table S2 is now included in the main manuscript and renamed to Table 1.

Please correct the 5XFAD label in figure 4C comparing astrocyte function. Response: corrected.

Please correct the typo in Nanostring in the caption of Figure 1B. Response: corrected.

We would like to thank reviewers once again for their insightful suggestions and interest in our work.

Reviewer 2 Report

Review of Makarava, et al.,

Neurodegenerative diseases share many characteristics, including being late onset and triggered by misfolding of specific proteins. Remarkably, these proteins are expressed throughout life, which begs the question of why disease takes so long to set in. More remarkably, the proteins are widely expressed yet target specific brain regions, a phenomenon known as selective vulnerability. A better understanding of the nature of selective vulnerability may improve our prospects of finding sorely needed therapies.

The authors make a good case for the need to understand what astrocytes are doing in response to challenges that induce astrocytosis.  They and others have previously reported that astrocytes have gene expression patterns that are strongly impacted by the region in which they reside. In the new study, they address the question of do astrocytes in specific regions always respond with the same gene expression change, regardless of the trigger. An important strength of the study is the use of many diverse challenges that induce astrocytosis. Makarava et al., demonstrate that these region-specific patterns are maintained during astrocytosis induced by 5 different insults, leaving an imprint on the response. This study provides an important contribution to addressing the problem of selective vulnerability. The authors do a great job interpreting the results and putting them into context. This is a very rich data set that will be a widely used resource for researchers interested in either astrocyte biology or selective vulnerability.

While I think this is already a very nice manuscript, I offer below some questions and comments that I hope the authors find useful as they revise their manuscript. 

1) A brief description of technical details on how the Nanostring method works would be useful. For example, is each sample measured once or multiple times? How much RNA is required? This can be an opportunity to explain the benefits of this method over others, such as QPCR and RNAseq.

Also, readers that are familiar with gene expression methods may wonder why not apply single-cell sequencing methods. The discussion section nicely describes the problem of losing mRNAs that are translated in astrocytic processes. And of course, it would be expensive, and probably technically not feasible to study 5 conditions in 4 regions as done here. However, another (bigger?) problem I see is that scRNAseq is dependent on cell identities. If during astrocytosis a cell changes the expression of cell identity markers, that cell becomes wrongly assigned to be a different cell type even though it would be an important result. Do you see that as being a problem for using scRNAseq to study how cells respond to disease?

2) Some more information on how the probes were selected and categorized would be helpful, in addition to the information provided in the methods and results sections. For example, when looking at Table S3 the gene ARNTL at the top caught my eye. ARNTL (AKA BMAL1, associated with CLOCK) was in the category  “Regulators of astrocyte reactivity|inflammation|CNS immunity”. While I am aware that mice with brain-specific ARNTL knocked out have a strong astrocytosis, if I was categorizing this gene this is probably not where I would put it. ARNTL is a widely expressed circadian rhythm gene, is not astrocyte-enriched, and in fact, is more highly expressed in neurons than astrocytes. Therefore, how was the list of genes established and categorized? Many of the genes in the links to online resources could be linked to multiple gene sets or pathways. How was this handled here? Was it from an in-house literature search, or something else? Just a few more sentences to include these details would be useful. Also, although with some effort a reader can sort Table S3 to find out what genes belong to each category, most readers will not make that extra effort. The authors would improve their manuscript by making this information easier to inspect as a smaller table or list.

3) The Nanostring library was used extensively in the authors’ previous articles. It is probably mentioned in them how the library was validated but would be good to summarize these details in the methods section. For example, a brief note about how the probes are validated to be specific to the intended gene. Perhaps this is something the company did.

4) Although two models of protein-misfolding-induced neurodegeneration were studied, the disease stages were very different. The prion mice were terminally ill while the AD mice were far from dying. A comparison of two models at comparable disease stages would be informative. The authors used this panel previously to analyze multiple stages of prion disease. It would be helpful if the authors could discuss, in the discussion section, how those previous results relate to these current results, especially regarding the AD model.

5) To me, the most important figure in the manuscript is Figure 3. It nicely demonstrates the overall finding. I have a simple suggestion to improve it. In Fig 3A the two panels could be combined into a single panel. For example, for the current left, the 6 tiles on the top right could be replaced with the corresponding tiles from the right panel. The result is that the same information is not displayed twice, and because space is conserved, the fused panel can be enlarged so it is easier to see the individual symbols.

6) In the abstract it is written: “Regardless of the nature of the insult or the insult-specificity of astrocyte response, strong correlations between the degree of astrocyte reactivity and perturbations in their homeostasis-associated genes were observed within each individual brain region. “

How is astrocyte reactivity quantified? Is it limited to the analysis in Figure 2? Immunofluorescence was done, could those analyses be connected to the expression analyses? Could this be a more sensitive way to quantify/score astrocytosis than a standard histological analysis? Was microglial activation also quantified?

Author Response

We sincerely appreciate the reviewers for their time and effort dedicated to reviewing our manuscript. We thank the reviewers for acknowledging the significance of our research and for their constructive feedback that has been invaluable for improving the quality and clarity of our research. All changes in the revised manuscript are tracked.

1) A brief description of technical details on how the Nanostring method works would be useful. For example, is each sample measured once or multiple times? How much RNA is required? This can be an opportunity to explain the benefits of this method over others, such as QPCR and RNAseq. Also, readers that are familiar with gene expression methods may wonder why not apply single-cell sequencing methods. The discussion section nicely describes the problem of losing mRNAs that are translated in astrocytic processes. And of course, it would be expensive, and probably technically not feasible to study 5 conditions in 4 regions as done here. However, another (bigger?) problem I see is that scRNAseq is dependent on cell identities. If during astrocytosis a cell changes the expression of cell identity markers, that cell becomes wrongly assigned to be a different cell type even though it would be an important result. Do you see that as being a problem for using scRNAseq to study how cells respond to disease?  

Response. We sincerely appreciate the reviewer for their insightful comments, valuable recommendations, and suggestions. In response to their feedback, we have expanded the description of the Nanostring approach within the Materials and Methods section to incorporate the specific details they had requested. Furthermore, we extend our gratitude for the additional inquiries related to the limitations of scRNAseq. We have incorporated these queries into the discussion. These modifications include integrating statements from the reviewer's comments concerning changes in expression within cell identity markers. We recognize the potential of the targeted gene analysis conducted through Nanostring, particularly its capacity to facilitate the comparison of a substantial number of samples across varying conditions. We view this technique as a valuable complement to scRNAseq. Given the ascendancy of scRNAseq as a prominent method in recent years, we are mindful not to alienate the numerous experts in the field who exclusively endorse scRNAseq. Our aim is to acknowledge the diverse perspectives within the scientific community and emphasize the synergistic value that both Nanostring and scRNAseq bring to advancing our understanding of cellular mechanisms.

2) Some more information on how the probes were selected and categorized would be helpful, in addition to the information provided in the methods and results sections. For example, when looking at Table S3 the gene ARNTL at the top caught my eye. ARNTL (AKA BMAL1, associated with CLOCK) was in the category  “Regulators of astrocyte reactivity|inflammation|CNS immunity”. While I am aware that mice with brain-specific ARNTL knocked out have a strong astrocytosis, if I was categorizing this gene this is probably not where I would put it. ARNTL is a widely expressed circadian rhythm gene, is not astrocyte-enriched, and in fact, is more highly expressed in neurons than astrocytes. Therefore, how was the list of genes established and categorized? Many of the genes in the links to online resources could be linked to multiple gene sets or pathways. How was this handled here? Was it from an in-house literature search, or something else? Just a few more sentences to include these details would be useful. Also, although with some effort a reader can sort Table S3 to find out what genes belong to each category, most readers will not make that extra effort. The authors would improve their manuscript by making this information easier to inspect as a smaller table or list.

Response. We thank the reviewer for raising this question, as we are firmly committed to the clarity of data presentation. In the process of selecting genes for the Astrocyte-specific panel, we utilized the compilation of astrocyte-enriched genes published by Network Glia, accessible at (https://www.networkglia.eu/en/astrocyte). To ascertain their astrocyte-specificity, we cross-referenced these genes with publicly available database www.brainrnaseq.org. For the validation of functional pathways, an in-house literature search was conducted. Among 47 genes reporting on reactive phenotypes were A1-, A2-, pan-specific markers, originally identified by Ben Barres and coauthors. We are aware that some of the A1-, A2- and pan-specific markers lacked astrocyte-specificity. Nevertheless, they were included in the panel as a part of broader list of astrocyte reactivity markers, because they have been widely utilized within the field for an extended period to monitor the nature of astrocyte reactivity. The revised Design of the Nanostring panel now incorporates this contextual information. Recognizing the constraint in the number of genes accommodated within the panels, we choose not to create an exhaustive listing of pathways encompassing all astrocyte functions. In certain instances, gene sets containing a limited number of genes were dissolved and their constituent genes were reallocated among other pathways. This strategic approach serves to mitigate the potential for misleading outcomes stemming from an overrepresentation of gene sets with a small number of genes. It's worth noting that during the panel's design in 2018, despite being recognized as a regulator of circadian rhythm, ARNTL (BMAL1) was incorporated into the gene set pertaining to astrocyte reactivity. This decision was predicated on a study indicating that the deletion of ARNTL induces astrocyte activation and prompts the expression of inflammatory genes (as documented in doi: 10.1016/j.celrep.2018.09.015). At the time, data from www.brainrnaseq.org indicated a predominant astrocyte-specific expression for ARNTL (BMAL1). We acknowledge that the information regarding cell-specific expression might have evolved over time. In the revised Table S2, a dedicated page (#5) has been introduced, where all genes are systematically sorted based on their corresponding Gene Set.

3) The Nanostring library was used extensively in the authors’ previous articles. It is probably mentioned in them how the library was validated but would be good to summarize these details in the methods section. For example, a brief note about how the probes are validated to be specific to the intended gene. Perhaps this is something the company did. \

Response. We value this insightful comment, as it provides us with the opportunity to enhance the description of the Nanostring method. The respective section has undergone a thorough revision, incorporating additional details. This includes a statement addressing the validation of the panel.

4) Although two models of protein-misfolding-induced neurodegeneration were studied, the disease stages were very different. The prion mice were terminally ill while the AD mice were far from dying. A comparison of two models at comparable disease stages would be informative. The authors used this panel previously to analyze multiple stages of prion disease. It would be helpful if the authors could discuss, in the discussion section, how those previous results relate to these current results, especially regarding the AD model.

Response. These are very good questions. We are not sure whether 5XFAD reach terminal stage and how such stage should be defined. 5XFADs develops amyloid plaques by 4 months of age and a full spectrum of pathological changes by 6-7 months of age. After that, these animals endure pathological changes well for at least a year in our lab without exhibiting substantial weight loss or other severe indicators necessitating euthanasia. Comparable findings are also reported in the literature (doi.org/10.1038/s41597-021-01054-y). Our rational for using 10-month old 5XFADs was to examine animals with fully developed pathology while avoiding an overlap between disease-specific and age-related changes. Unfortunately, our prior investigations into various stages of prion disease employed distinct Nanostring panels (specifically the Neuroinflammation panel) developed by Nanostring. Due to substantial dissimilarities in gene composition, employing statistical methods to compare outcomes derived from two distinct panels is inappropriate. In our revised discussion, we indicate that the lower scores in the 5XFAD model relative to the prion-infected mice could be attributed to several factors: (i) inherent milder disease pathology in the 5XFAD model compared to prions. (ii) a disease stage that is less advanced in the 5XFAD model; (iii) the observation that in 5XFAD mice, astrocytes are activated in only close proximity to Ab plaques, and are diluted with homeostatic astrocytes in bulk tissue analysis.

5) To me, the most important figure in the manuscript is Figure 3. It nicely demonstrates the overall finding. I have a simple suggestion to improve it. In Fig 3A the two panels could be combined into a single panel. For example, for the current left, the 6 tiles on the top right could be replaced with the corresponding tiles from the right panel. The result is that the same information is not displayed twice, and because space is conserved, the fused panel can be enlarged so it is easier to see the individual symbols.

Response. We thank the reviewer for this valuable suggestion. The revised the Figure 3 exactly as suggested in this comment.

6) In the abstract it is written: “Regardless of the nature of the insult or the insult-specificity of astrocyte response, strong correlations between the degree of astrocyte reactivity and perturbations in their homeostasis-associated genes were observed within each individual brain region. “ How is astrocyte reactivity quantified? Is it limited to the analysis in Figure 2? Immunofluorescence was done, could those analyses be connected to the expression analyses? Could this be a more sensitive way to quantify/score astrocytosis than a standard histological analysis? Was microglial activation also quantified?

 Response. Due to the specific focus of the current study on astrocyte reactivity, microglia reactivity was not quantified. However, it is worth noting that including additional genes reporting on the reactive state of microglia could enhance the robustness of quantifications. For quantifying astrocyte reactivity, GSA scores, which reflects a number of DEGs, weighted differences in their fold change and statistical significance, were calculated using Nanostring nCounter built-in algorithms for 47 genes included in 4 astrocyte reactivity gene sets. In a similar manner, for assessing astrocytic functional changes, GSA scores were calculated for other gene set encompassing 275 genes that expressed predominantly in astrocytes and report on their homeostatic functions. These statements are now included in corresponding chapter “The degree of astrocyte reactivity correlates …”. Traditionally, the histological analysis of astrocyte reactivity has relied on the quantification of a limited number of markers such as GFAP and/or Aldh1l1. However, we believe that this approach may lack precision since these markers might not fully capture the complexity and extent of astrocyte transformation. In a previous investigation, we examined region-specific astrocyte reactivity in response to prion infection using immunohistochemistry with GFAP, S100b, and Aldh1l1 markers (DOI.org/10.3389/fnins.2019.01048). Notably, even though prion infection severely impacts the thalamus, the reactive astrocytes in this region exhibited less GFAP reactivity compared to astrocytes in the cortex and hippocampus. Our previous work convincingly demonstrated that astrocyte responses to prions are contextually specific to different brain regions. This insight served as an inspiration for the present study, which seeks to determine whether the region-specific response is influenced by the nature of the insult. A brief discussion comparing the assessment of astrocyte reactivity using gene expression analysis and traditional immunostaining has been included in the Discussion section.

We would like to thank reviewers once again for their insightful suggestions and interest in our work.

Reviewer 3 Report

The manuscript by Makarava et al. presents an analysis of gene expression in different brain regions and different mouse models of neurological insult and disease, including aging, Alzheimer’s disease, brain ischemia, traumatic brain injury and prion disease. The authors employed a targeted analysis of gene expression in bulk tissues, focusing on 275 genes that are thought to be primarily expressed by astrocytes under normal conditions and 47 markers that are thought to reflect reactive astrocytes. The study was designed to assess astrocyte-associated gene expression across neuropathological insults and brain regions, which is of interest to other researchers when comparing transcriptional datasets between tissues and disease conditions.

Solely based on a targeted analysis of gene expression in bulk brain tissues, the authors make a case that brain regional variation is greater than insult-specific variation when examining gene expression associated with astrocytes. This narrative based on different reactive astrocyte states/phenotypes/identities perhaps overreaches the conclusions presented because bulk gene expression alone is unable to distinguish between astrocyte states, phenotypes, or subtypes. Revisions would be beneficial prior to publication.

Major comments

1)

The manuscript claims to have resolved the region-specific identity of reactive astrocytes, which implies that astrocytes are qualitatively different between brain regions. However, it is unclear how the authors can distinguish quantitative versus qualitative differences in astrocyte reactivity. For instance, certain genes might not reach statistical significance until astrocytes reach a certain quantitative level of reactivity. In supplementary figure 1, immunohistochemistry of microglia and astrocytes in prion-infected and Alzheimer’s disease mice is presented qualitatively. However, a quantitative assessment of astrocyte reactivity (separate from gene expression) across all brain regions and neuropathological insults is needed to distinguish whether regional differences in reactive astrocyte marker gene expression reflects the quantity or quality/phenotype of reactive astrocytes.

2)

Related to the previous comment, the claim of brain-region-specific astrocyte identities requires more than a simple analysis of average gene expression in bulk tissues. The average gene expression measured in this study was a function of the sub-populations of different brain cell types that differ between different brain regions. While the genes investigated might be primarily expressed by astrocytes, other types of brain cells likely make some contribution to the average of gene expression measured, which likely differs based on brain region. Without an analysis of gene expression in individual astrocytes, the authors cannot conclude whether the observed gene expression changes were because reactive astrocytes are different between brain regions (i.e. region-specific identities), or just because the baseline average of gene expression differs based on brain region.

3)

Line 306-307: “The degree of astrocyte reactivity correlates with the degree of changes in expression of genes associated with astrocyte functions.”

This conclusion is not necessarily valid because the degree of astrocyte reactivity was not actually measured. In figure 2, the authors attempt to correlate expression of genes that reflect “astrocyte reactivity” with genes that reflect “physiological functions”. As gene expression was measured in bulk tissues, it’s not actually possible for the authors to conclude which genes specifically originated from reactive astrocytes. Furthermore, the observed correlation was likely because reactive astrocytes can proliferate in association with neurological disease/insult, causing all astrocyte-associated genes to change proportionally with the level of proliferation. In other words, this correlation is likely merely a function of biological effect size. To support the conclusion cellular composition must be measured independently in different conditions through a quantitative histology-based approach for correlation. (i.e. “These results suggest that regardless of the nature of the insult or the identity of astrocyte reactive phenotype, the changes in functional pathways are tightly linked to and perhaps determined by the degree of astrocyte reactivity.”) Analysis of gene expression within individual astrocytes, or at the very least gene expression specifically within astrocytes but not other brain cell types would further strengthen the conclusion.

4)

Lines 326-358: “Region-specific continuums of astrocyte reactive states.”

The conclusion was based solely on Principle Component Analysis (PCA) of bulk-tissue gene expression, which cannot distinguish between cell types and is merely a means of measuring the overall inter-sample variation in gene expression. Therefore, the “region-specific” variation in gene expression could be because the baseline gene expression differs based on brain region, not because reactive astrocytes are actually different between brain regions. Non-astrocyte genes were included for the PCA analysis shown in Figure 3A, which leads to some bias in the analysis of variation within astrocyte-associated gene expression (Line 326). Furthermore, in Figure 3B, a PCA of microglial and neuronal gene expression is presented for different brain regions and pathogenic insults. However, the 8 microglial genes and 10 neuronal genes included in the panel are insufficient for this type of analysis. The conclusions pertaining to microglia and neurons are therefore not robust. Greater separation could likely be achieved by including more microglial and neuronal genes in the analysis. As previously stated unbiased gene expression measurements in specific cell types would be required to make robust conclusions on the region-specific and insult-specific variation in gene expression of different cell types and states.

5)

Line 359: “Astrocyte remodeling involves a core gene set common across insults.”

Functional studies or cell-type-specific measurements of gene expression would be required to conclude that the genes are involved in astrocyte remodeling and so maybe this is perhaps too strong of a conclusion. The authors also focused on the cortex because “the degree of response in other brain regions appeared to be more dependent on the nature of the insult”. This may bias the results towards finding a core set of genes that were common across insults. In addition, this “core set” of astrocyte genes included microglia markers (CD68 and TLR2), which obviously are not involved in astrocyte remodeling.

6)

Line 393:427: “Insult-specific changes in astrocyte phenotypes.”

In this section, the authors discuss differences in astrocyte phenotypes across insults, primarily based on the heatmaps of log2-fold-change values presented in Figure 5. However, they cannot distinguish astrocyte phenotypes solely based on bulk-tissue gene expression. They also claim that “astrocyte homeostatic genes” (Figure 5B) display greater insult-specificity compared to “reactive astrocyte genes” (Figure 5A), yet to me both heatmaps appeared to display a similar level of insult-specificity. Furthermore, many of genes presented in Figure 5 are not expressed by astrocytes primarily (e.g. H2-D1, Cd14, Csf1, Serping1, H2-T23, Gad1, Bdnf etc.). This conclusion (and the next conclusion at lines 435:443) would again seem to require single-cell resolution, or validation by an orthogonal technique that can distinguish astrocyte phenotypes.

7)

Line 186: “Only genes with adjusted p<0.1, and linear fold change ≥±1.2 were counted.”

This is relaxed criteria for differential expression, which could artificially inflate the number of gene expression changes in various conditions. What is the rationale for such a relaxed cut-off? Are similar results obtained with standard criteria for differential expression? (e.g. adjusted p<0.05, and linear fold change ≥±1.5).

Minor comments

Lines 47-49: Both microglia and astrocytes are able to respond to a variety of pathological conditions.

Lines 101-102: Maybe here you could mention the timepoint did that prion-infected mice reached clinical endpoint for this study.

Lines 571-572: It is not clear whether astrocyte gene expression “drives” astrocyte reactivity, or whether astrocyte reactivity “drives” gene expression changes.

Author Response

We sincerely appreciate the reviewers for their time and effort dedicated to reviewing our manuscript. All changes in the revised manuscript are tracked.

Major comments

1.The manuscript claims to have resolved the region-specific identity of reactive astrocytes, which implies that astrocytes are qualitatively different between brain regions. However, it is unclear how the authors can distinguish quantitative versus qualitative differences in astrocyte reactivity. For instance, certain genes might not reach statistical significance until astrocytes reach a certain quantitative level of reactivity. In supplementary figure 1, immunohistochemistry of microglia and astrocytes in prion-infected and Alzheimer’s disease mice is presented qualitatively. However, a quantitative assessment of astrocyte reactivity (separate from gene expression) across all brain regions and neuropathological insults is needed to distinguish whether regional differences in reactive astrocyte marker gene expression reflects the quantity or quality/phenotype of reactive astrocytes. 

Response: We express our gratitude to the reviewer for raising this insightful comment, which provides us with an opportunity to clarify our findings. We apologize for the perception that our studies resolved the region-specific identity, which appears to be due to misunderstanding. Our results do, indeed, harmonize excellently with the current field's consensus that, under normal conditions, astrocytes do exhibit region-specific identities. Regarding reactive astrocytes, our intention was never to assert region-specific identities. We have revised the manuscript to eliminate any phrasing that might lead to the inference that we have definitively established the region-specific identity of reactive astrocytes. The term "identity" is now employed in relation to other studies. We have also made adjustments to the title to reflect these clarifications. Addressing the query about quantitative versus qualitative differences in reactive astrocytes, our preceding research, which inspired this current study, addressed this question. Our prior manuscript utilized immunohistochemistry along with gene expression analysis through RT-PCR to demonstrate that the astrocytic response to prion infection is defined by the brain region. While our approach in the current work does not possess the resolution required to conclusively ascertain region-specific identity, we are pleased to observe that our discoveries align congruently with studies implementing single-cell transcriptomics. Notably, a presentation by Dr. De Pitta (University of Toronto) during the Glia2023 conference scrutinized disease-specific astrocyte gene signatures in the context of the spatial and temporal dimensions of Alzheimer's disease progression. This presentation concluded that alterations specific to astrocytes in Alzheimer's disease are indeed delineated by region. A 2020 Nature Neuroscience publication authored by Habib et al. reported that a subpopulation of astrocytes in aged wild type mice and human brain were  phenotypically similar to a subpopulation of disease-associated astrocytes identified in the 5XFAD AD mouse model (Habib et al., 2020). We also observed overlap in clusters corresponding to the same mouse model of AD and 24-month-old aged wild type mice.

2.Related to the previous comment, the claim of brain-region-specific astrocyte identities requires more than a simple analysis of average gene expression in bulk tissues. The average gene expression measured in this study was a function of the sub-populations of different brain cell types that differ between different brain regions. While the genes investigated might be primarily expressed by astrocytes, other types of brain cells likely make some contribution to the average of gene expression measured, which likely differs based on brain region. Without an analysis of gene expression in individual astrocytes, the authors cannot conclude whether the observed gene expression changes were because reactive astrocytes are different between brain regions (i.e. region-specific identities), or just because the baseline average of gene expression differs based on brain region.

Response: We greatly value your perceptive insights regarding the clarity of our results and observations. In response to the earlier feedback, we have diligently revised the manuscript, ensuring the removal of any phrasing that could potentially imply the establishment of region-specific identity in reactive astrocytes. Furthermore, we have refined the concluding segments of the discussion to offer a more comprehensive assessment of the inherent limitations inherent in the present study. Notably, we have incorporated a statement from the reviewers' commentary concerning the role of other cell types. It is undeniable that various cell types contribute to the overall gene expression profile that we have scrutinized. However, it's essential to recognize that cells retain their distinctive cell-specific identities even under pathological circumstances. In other words, astrocytes do not undergo a transformation into microglia, and vice versa. Because our approach involve monitoring gene sets, not individual genes, we believe our approach reflects changes predominantly occurring in astrocytes, not other cell types. Moreover, we performed principle component analysis not only for the whole panel (Fig. 3A), but for the several subsets of genes related to various astrocyte functions (Fig. 3B). For example, the continuum of changes within each brain region was observed for the gene set of transporters that were strictly selected based on their predominant expression in astrocytes. As mentioned in our discussion, we believe that the methodology relying on bulk tissue analysis complements the single-cell approach. While the latter provides exceptional resolution at the individual cell level, it is vital to acknowledge its inherent limitations. As our understanding grows regarding astrocytes' ability to translate mRNAs connected with their biological functions within their peripheral processes, the single-cell approach might inadvertently misrepresent the transcript count that reflects astrocytic functions.

3.Line 306-307: “The degree of astrocyte reactivity correlates with the degree of changes in expression of genes associated with astrocyte functions.”This conclusion is not necessarily valid because the degree of astrocyte reactivity was not actually measured. In figure 2, the authors attempt to correlate expression of genes that reflect “astrocyte reactivity” with genes that reflect “physiological functions”. As gene expression was measured in bulk tissues, it’s not actually possible for the authors to conclude which genes specifically originated from reactive astrocytes. Furthermore, the observed correlation was likely because reactive astrocytes can proliferate in association with neurological disease/insult, causing all astrocyte-associated genes to change proportionally with the level of proliferation. In other words, this correlation is likely merely a function of biological effect size. To support the conclusion cellular composition must be measured independently in different conditions through a quantitative histology-based approach for correlation. (i.e. “These results suggest that regardless of the nature of the insult or the identity of astrocyte reactive phenotype, the changes in functional pathways are tightly linked to and perhaps determined by the degree of astrocyte reactivity.”) Analysis of gene expression within individual astrocytes, or at the very least gene expression specifically within astrocytes but not other brain cell types would further strengthen the conclusion.

Response: We extend our gratitude to the reviewer for affording us the opportunity to delve deeper into this subject. The presented comment operates under the assumption that the proliferation of reactive astrocytes is a prevalent trait among various neurological conditions. Thus, the observed correlation likely stems from the sheer biological size. We genuinely value the insightful perspective offered by the reviewer, which we have thoroughly pondered upon. Although reactive astrocytes do undergo proliferation in response to certain insults such as scar formation, this phenomenon is not witnessed in instances of prion diseases or AD mouse models. In fact, the lack of astrocyte proliferation in prion disease is well established, yet prion-infected mice showed the highest GSA scores. Ribosomal profiling of different cell types and analysis of single-cell transcriptome revealed that a number of astrocyte do not change or might even decline  in prion-infected mice, (references:  doi.org/10.7554/eLife.62911; doi.org/10.1186/s40478-022-01450-4). Furthermore, when examining plaque-forming AD mouse models, a scrutiny of cell proliferation patterns revealed the proliferation of microglia, but lack of such for astrocytes (reference: doi.org/10.1002/glia.22295). Anticipating significant astrocyte proliferation within a 24-hour timeframe following MCAO is challenging. In cases of TBI, astrocyte proliferation primarily occurs at the site of the primary injury (reference: doi.org/10.1016/j.expneurol.2015.03.020). Notably, even though the cortex served as the principal site of injury, the thalamus displayed the highest GSA scores in relation to both astrocyte reactivity and function within the TBI group in the current work. Were the correlation solely attributable to astrocyte proliferation – in other words, the biological size – one would anticipate such a connection exclusively within the primary injury sites, namely the cortex. Intriguingly, the correlation was observable across all four brain regions, irrespective of the expected astrocyte proliferation in the primary lesion sites. This observation suggests that proliferation is not the predominant factor underpinning this correlation. We delve further into this topic within the Discussion, introducing the possibility of the proliferation hypothesis.

4.Lines 326-358: “Region-specific continuums of astrocyte reactive states. ”The conclusion was based solely on Principle Component Analysis (PCA) of bulk-tissue gene expression, which cannot distinguish between cell types and is merely a means of measuring the overall inter-sample variation in gene expression. Therefore, the “region-specific” variation in gene expression could be because the baseline gene expression differs based on brain region, not because reactive astrocytes are actually different between brain regions. Non-astrocyte genes were included for the PCA analysis shown in Figure 3A, which leads to some bias in the analysis of variation within astrocyte-associated gene expression (Line 326). Furthermore, in Figure 3B, a PCA of microglial and neuronal gene expression is presented for different brain regions and pathogenic insults. However, the 8 microglial genes and 10 neuronal genes included in the panel are insufficient for this type of analysis. The conclusions pertaining to microglia and neurons are therefore not robust. Greater separation could likely be achieved by including more microglial and neuronal genes in the analysis. As previously stated unbiased gene expression measurements in specific cell types would be required to make robust conclusions on the region-specific and insult-specific variation in gene expression of different cell types and states.

Response: We express our gratitude to the reviewer for raising this question, which provides us with the opportunity to clarify our findings. Our preceding investigation unveiled significant, region-specific differences in the astrocytic response to prion infection. This was assessed through an examination of cell morphology, subcellular localization, immunofluorescence intensity, and gene expression patterns of GFAP, S100b, and Aldh1l1, utilizing immunohistochemistry and qRT-PCR techniques, respectively (DOI.org/10.3389/fnins.2019.01048). In contrast to the thalamus, where the changes in astrocyte morphology were subtle in response to prions, reactive astrocytes within the cortex and hippocampus exhibited significantly enlarged cell bodies and thickened processes. The region-specific differences in the morphology of reactive astrocytes and their immunofluorescence patterns align harmoniously with the PCA results presented in Figure 3 of our current study. These findings underscore the similarity between the cortical and hippocampal continuums, while starkly differing from the thalamic continuum. The revised text omits any references to the reactive state of astrocytes. The discussion section was revised to include comparison of the current data with histopathological analysis.

We agree that more microglial and neuronal genes are needed for a stronger principal component analysis on microglia and neurons. Interestingly, 10 genes representing neuronal function were sufficient to illustrate the well-known region-specificity of neurons. Concerning the insult-specific separation of neuronal genes, we modified our statement to add clarity. To improve PCA of microglial genes, we increased the Microglia gene set by including 5 additional genes into the analysis: Ccl2, Csf1 and Socs3, which according to the Brain RNA-Seq database are expressed in microglia, and Apoe and Vegfa that are characteristic of the disease-associated microglia (DAM). The Fig. 3B was updated with a new PCA for microglia.

5.Line 359: “Astrocyte remodeling involves a core gene set common across insults.” Functional studies or cell-type-specific measurements of gene expression would be required to conclude that the genes are involved in astrocyte remodeling and so maybe this is perhaps too strong of a conclusion. The authors also focused on the cortex because “the degree of response in other brain regions appeared to be more dependent on the nature of the insult”. This may bias the results towards finding a core set of genes that were common across insults. In addition, this “core set” of astrocyte genes included microglia markers (CD68 and TLR2), which obviously are not involved in astrocyte remodeling.

Response: We agree with this comment. In revised manuscript, the sub-chapter ““Astrocyte remodeling involves a core gene set common across insults” is now merged with the following chapter “Insult-specific changes…”. We revised and significantly reduced the paragraph describing common genes shared between insults, softened the language, removed references to “core set” or “core gene set”, and removed the statement regarding the “core gene set” from the Abstract.

6.Line 393:427: “Insult-specific changes in astrocyte phenotypes.” In this section, the authors discuss differences in astrocyte phenotypes across insults, primarily based on the heatmaps of log2-fold-change values presented in Figure 5. However, they cannot distinguish astrocyte phenotypes solely based on bulk-tissue gene expression. They also claim that “astrocyte homeostatic genes” (Figure 5B) display greater insult-specificity compared to “reactive astrocyte genes” (Figure 5A), yet to me both heatmaps appeared to display a similar level of insult-specificity. Furthermore, many of genes presented in Figure 5 are not expressed by astrocytes primarily (e.g. H2-D1, Cd14, Csf1, Serping1, H2-T23, Gad1, Bdnf etc.). This conclusion (and the next conclusion at lines 435:443) would again seem to require single-cell resolution, or validation by an orthogonal technique that can distinguish astrocyte phenotypes.

Response: In response to this comment, we revised the language and removed any references to astrocyte phenotype. We also removed the claim that “astrocyte homeostatic genes” (Figure 5B) display greater insult-specificity compared to “reactive astrocyte gene”. For assessing astrocyte reactivity, A1-, A2- and pan-specific markers originally identified by Ben Barres along with other markers of astrocyte reactivity expressed by astrocytes were included in the gene panel. It is important to note that while some of the markers associated with A1, A2, and pan-specific reactions from Barres' work may not exclusively pertain to astrocytes, they have been widely utilized within the field for an extended period to monitor the nature of astrocyte reactivity. Although acknowledging the non-astrocyte specificity of certain A1, A2, and pan-specific markers from Barres' research, we made the decision to retain these markers due to their established history as reference points for gauging astrocyte reactivity. At the time this study was initiated in 2018, the perceptions of A1, A2, and pan-reactive astrocytes within the field were distinct from the current understanding. With regard to the design of the Nanostring panel, the Method section has been revised to explicitly outline that A1, A2, and pan-reactive markers are not expressed exclusively by astrocytes.

7.Line 186: “Only genes with adjusted p<0.1, and linear fold change ≥±1.2 were counted.”This is relaxed criteria for differential expression, which could artificially inflate the number of gene expression changes in various conditions. What is the rationale for such a relaxed cut-off? Are similar results obtained with standard criteria for differential expression? (e.g. adjusted p<0.05, and linear fold change ≥±1.5).

Response: We thank the reviewer for this question that allows to clarify the rational behind adjusted p<0.05. We experimented with different cut-offs and found out that due to the low number of samples in some groups (MCAO, TBI), more stringent criteria can result in underreporting of important DEGs. For example, GFAP changes in contralateral thalamus with TBI are detected with adjusted p=0.0541. However, with linear fold change of 2.74 and non-adjusted p=0.0000533, GFAP is most likely to be differentially expressed. More examples of DEGs that would be lost with p<0.05 instead of p<0.1 cut-off are listed in the attachment. Notably, vast majority of DEGs with adjusted p<0.1 have non-adjusted p value <0.01 (see Table S3). Using more stringent criteria for linear fold change does result in less DEGs in all groups but does not change the overall trends (see example for using different criteria in cortex, attached).

 Minor comments

Lines 47-49: Both microglia and astrocytes are able to respond to a variety of pathological conditions. Response: It is true, this sentence is revised and the reference to microglia is removed.

Lines 101-102: Maybe here you could mention the timepoint did that prion-infected mice reached clinical endpoint for this study. Response: This information is provided in Table 1, which is now included in main text.

Lines 571-572: It is not clear whether astrocyte gene expression “drives” astrocyte reactivity, or whether astrocyte reactivity “drives” gene expression changes. Response: This sentence is removed.

We would like to thank reviewers once again for their insightful suggestions and interest in our work.
